# A novel antifolate suppresses growth of FPGS-deficient cells and overcomes methotrexate resistance

Felix van der Krift[1], Dick W Zijlmans[2], Rhythm Shukla[3], Ali Javed[3,10], Panagiotis I Koukos[4], Laura LE Schwarz[1], Elpetra PM Timmermans-Sprang[5], Peter EM Maas[6], Digvijay Gahtory[7], Maurits van den Nieuwboer[7], Jan A Mol[5], Ger J Strous[8], Alexandre MJJ Bonvin[4], Mario van der Stelt[9], Edwin JA Veldhuizen[10], Markus Weingarth[3], Michiel Vermeulen[2], Judith Klumperman[8], Madelon M Maurice[1]

Cancer cells make extensive use of the folate cycle to sustain increased anabolic metabolism. Multiple chemotherapeutic drugs interfere with the folate cycle, including methotrexate and 5-fluorouracil that are commonly applied for the treatment of leukemia and colorectal cancer (CRC), respectively. Despite high success rates, therapy-induced resistance causes relapse at later disease stages. Depletion of folylpolyglutamate synthetase (FPGS), which normally promotes intracellular accumulation and activity of natural folates and methotrexate, is linked to methotrexate and 5-fluorouracil resistance and its association with relapse illustrates the need for improved intervention strategies. Here, we describe a novel antifolate (C1) that, like methotrexate, potently inhibits dihydrofolate reductase and downstream one-carbon metabolism. Contrary to methotrexate, C1 displays optimal efficacy in FPGS-deficient contexts, due to decreased competition with intracellular folates for interaction with dihydrofolate reductase. We show that FPGS-deficient patient-derived CRC organoids display enhanced sensitivity to C1, whereas FPGS-high CRC organoids are more sensitive to methotrexate. Our results argue that polyglutamylation-independent antifolates can be applied to exert selective pressure on FPGS-deficient cells during chemotherapy, using a vulnerability created by polyglutamylation deficiency.

## Introduction

Antifolates constitute a subclass of antimetabolites that have been applied as chemotherapeutic agents for decades (Wilson et al, 2014; Stine et al, 2022). In mammals, folate is an essential vitamin that functions as cofactor for the one-carbon cycle, which is among the most highly and specifically up-regulated pathways in cancer (Nilsson et al, 2014). The enzyme dihydrofolate reductase (DHFR) activates folates by reducing inactive, oxidized dihydrofolate (DHF) to tetrahydrofolate (THF), which is inter-converted to 5,10-methylene-THF, 10-formyl-THF, and 5-methyl-THF by various enzymes of the one-carbon cycle. THF functions as a carrier for serine- and glycine-derived one-carbon units and is indispensable for supplying substrates to enzymatic pathways required for de novo nucleotide synthesis, NAD(P)H generation, ATP synthesis, amino acid homeostasis, and tRNA modification (Yang & Vousden, 2016; Ducker & Rabinowitz, 2017; Zheng & Cantley, 2019). Folate metabolism is compartmentalized at the subcellular level, with similar reaction steps occurring in the cytosol and mitochondria, which is essential to maintain folate integrity (Zheng et al, 2018). Import of folate from the extra-cellular environment into the cytosol is mediated via the membrane-bound solute carriers SLC19A1 and SLC46A1 or via clathrin-mediated endocytosis of folate receptors (FOLR1-3) (Zhao et al, 2009, 2011; Zheng & Cantley, 2019). Once imported, folates are polyglutamylated by folylpolyglutamate synthetase (FPGS), which catalyzes the addition of negatively charged glutamate residues to prevent efflux and promote intracellular accumulation (Osborne et al, 1993; Lawrence et al, 2014). Because of the crucial role of DHFR in maintaining sufficient concen-trations of THF to drive anabolic metabolism, multiple DHFR inhibitors are clinically available for the treatment of neoplastic and autoimmune diseases, including the commonly used anti-folate methotrexate. Next to targeting DHFR, the one-carbon cycle is inhibited by drugs directed at thymidylate synthase

[1]Center for Molecular Medicine and Oncode Institute, University Medical Center Utrecht, Utrecht, The Netherlands   [2]Department of Molecular Biology and Oncode Institute, Faculty of Science, Radboud Institute for Molecular Life Sciences, Radboud University Nijmegen, Nijmegen, The Netherlands   [3]NMR Spectroscopy, Bijvoet Centre for Biomolecular Research, Department of Chemistry, Faculty of Science, Utrecht University, Utrecht, The Netherlands   [4]Computational Structural Biology, Bijvoet Centre for Biomolecular Research, Faculty of Science, Department of Chemistry, Utrecht University, Utrecht, The Netherlands   [5]Department of Clinical Sciences of Companion Animals, Utrecht University, Utrecht, The Netherlands   [6]Specs Compound Handling B.V., Zoetermeer, The Netherlands   [7]BIMINI Biotech B.V., Leiden, The Netherlands   [8]Center for Molecular Medicine, Cell Biology, University Medical Center Utrecht, Utrecht, The Netherlands   [9]Department of Molecular Physiology and Oncode Institute, Leiden Institute of Chemistry, Leiden University, Leiden, The Netherlands   [10]Division of Infectious Diseases and Immunology, Department of Biomolecular Health Sciences, Utrecht University, Utrecht, The Netherlands

Correspondence: M.M.Maurice@umcutrecht.nl

(TYMS), like 5-fluorouracil (5-FU), which is used as first-line therapy for colorectal cancer (CRC) (Stine et al, 2022).

Methotrexate is widely used to treat tumors, often in combination with other chemotherapeutics (Stine et al, 2022). Despite the high success rate of methotrexate treatment, therapy resistance presents a major problem, for example, in pediatric acute lymphoblastic leukemia (ALL), where relapses occur in 20% of patients (Nguyen et al, 2008). Acquired chemoresistance observed in relapsed patients suggests that initial chemotherapy drives selection of drug-resistant clones (Li et al, 2020; Yu et al, 2020). At the cellular level, resistance to antifolates develops through various mechanisms that cause decreased cellular import, decreased retention, or increased export (Zhao & Goldman, 2003; Fotoohi et al, 2009; Zarou et al, 2021). Methotrexate is an FPGS substrate and its polyglutamylation is required for efficient intracellular retention and the selective targeting of tumor cells (Fabre et al, 1984; Rots et al, 1999). Studies in cell lines and cancer cells derived from relapsed patients showed that methotrexate-induced FPGS deficiency develops through transcriptional down-regulation or the acquirement of inactivating mutations (Liani et al, 2003; Fotoohi et al, 2009; Stark et al, 2009; Wojtuszkiewicz et al, 2016; Li et al, 2020; Yu et al, 2020). Two recent studies showed that at least 8% of relapsed ALL patients obtained FPGS mutations upon methotrexate therapy (Li et al, 2020; Yu et al, 2020). Moreover, the contribution of FPGS deficiency to methotrexate resistance may be underestimated, as over 50% of relapsed patients display transcriptional *FPGS* down-regulation (Li et al, 2020; Yu et al, 2020). Furthermore, FPGS deficiency has been linked to resistance to 5-FU in models of CRC (Sohn et al, 2004), illustrating the need for novel intervention strategies to prevent relapses caused by drug-resistant, polyglutamylation-deficient cells.

Here, we report on the mechanistic characterization of a 2,4-diaminopyrimidine–derivative that we called compound 1 (C1), a novel and highly potent, non-classical antifolate that targets FPGS-deficient cells. Functional comparison with methotrexate suggests that, although their cellular targets are similar, individual cancer cell lines display up to a 50-fold difference in sensitivity to both drugs. We demonstrate that cells with low FPGS expression are prone to DHFR inhibition by polyglutamylation-independent non-classical antifolates, such as C1 and pyrimethamine. Using patient-derived CRC organoids that display either FPGS deficiency or overexpression, we confirm that C1 selectively inhibits growth of FPGS-deficient cells, whereas cells with high FPGS expression display sensitivity to methotrexate, a classical antifolate. Our results show that polyglutamylation-independent non-classical antifolates like C1 exert selective pressure on FPGS-deficient cells during chemotherapy and thus may be applied to prevent tumor evolution towards a methotrexate-resistant subtype. In comparison with trimetrexate, the lipophilic and FPGS-independent derivative of methotrexate, C1 has increased potency towards polyglutamylation-deficient cells. We anticipate that C1's structure may serve as a template for development of improved non-classical antifolates.

# Results

## Identification of a 2,4-diaminopyrimidine–based compound as a novel DHFR ligand

In a screen of compounds with potential antineoplastic activity (Van der Velden et al, 2022), we identified a 2,4-diaminopyrimidine–derivative (C1, also referred to as BM001 [Van der Velden et al, 2022]) that selectively inhibited growth of a subset of cancer cell lines (Table S1). We noted that C1 shares structural features with non-classical antifolates like pyrimethamine (Fig 1A). As C1 is structurally divergent from the classical antifolate methotrexate (Fig 1A), we compared the growth inhibitory activity of both compounds towards a panel of cancer cell lines listed in the Cancer Cell Line Encyclopedia (Ghandi et al, 2019). Remarkably, $IC_{50}$ values between C1 and methotrexate did not correlate (Pearson r = 0.058), and a subset of cell lines even displayed up to 50-fold differences in sensitivity (Fig 1B). These results thus suggest that, despite its predicted antifolate activity, C1 inhibits tumor cell growth by a mode of action different from methotrexate.

To understand the underlying growth inhibitory mechanism, we aimed to identify key cellular proteins targeted by C1. Based on the presence of a nitrogenous base, we first hypothesized that C1 may act as an ATP-competitive kinase inhibitor. However, we did not detect in vitro kinase inhibition using 100 nM C1 in a kinome-wide screen (Fig S1). Next, we interrogated C1-protein interactions by applying mass spectrometry–based thermal proteome profiling (TPP) (Molina et al, 2013; Savitski et al, 2014; Mateus et al, 2020a, 2020b). We used the LS 174T CRC cell line, that is, sensitive to C1 treatment (Fig 1C). LS 174T cells were either mock-treated or treated for 1 h with 10 $\mu$M C1, after which cells were subjected to TPP. A remarkable C1-mediated thermal shift was observed for DHFR, indicating that C1 primarily acts as a DHFR ligand, as suspected, based on its 2,4-diaminopyrimidine group (Fig 1D). In addition, we observed C1-mediated stabilization for TYMS (Table S2), a known nM affinity target of polyglutamylated methotrexate (Huber et al, 2015). This finding was unexpected, as C1 lacks a glutamate group for FPGS-mediated polyglutamylation. We therefore speculate that TYMS may instead be stabilized by the accumulation of dUMP, due to 5,10-methylene-THF depletion that occurs in DHFR-inhibited cells (Yang & Vousden, 2016; Ducker & Rabinowitz, 2017; Zheng & Cantley, 2019; Brown et al, 2023).

To validate C1 as a novel DHFR ligand, we overexpressed human DHFR (hDHFR) bearing an N-terminal HA-tag in HEK293T cells and performed a cellular thermal shift assay (CETSA) after treatment for 1 h with either 10 $\mu$M C1 or methotrexate (Molina et al, 2013). Western blot analysis revealed that both methotrexate and C1 stabilize HA–DHFR when compared with DMSO-treated cells, whereas the thermal stability of control proteins (AKT and actin) remained unaffected by both drug treatments (Fig 1E). These findings thus confirm that treatment with C1, like methotrexate, stabilizes DHFR in live cells, suggesting a direct interaction. Based on our combined findings, we conclude that C1 is a novel DHFR ligand, similar to methotrexate. The observed differential growth inhibitory effects of C1 and methotrexate on various cancer cell lines, however, indicates that their mode of action does not fully overlap or may be context-dependent.

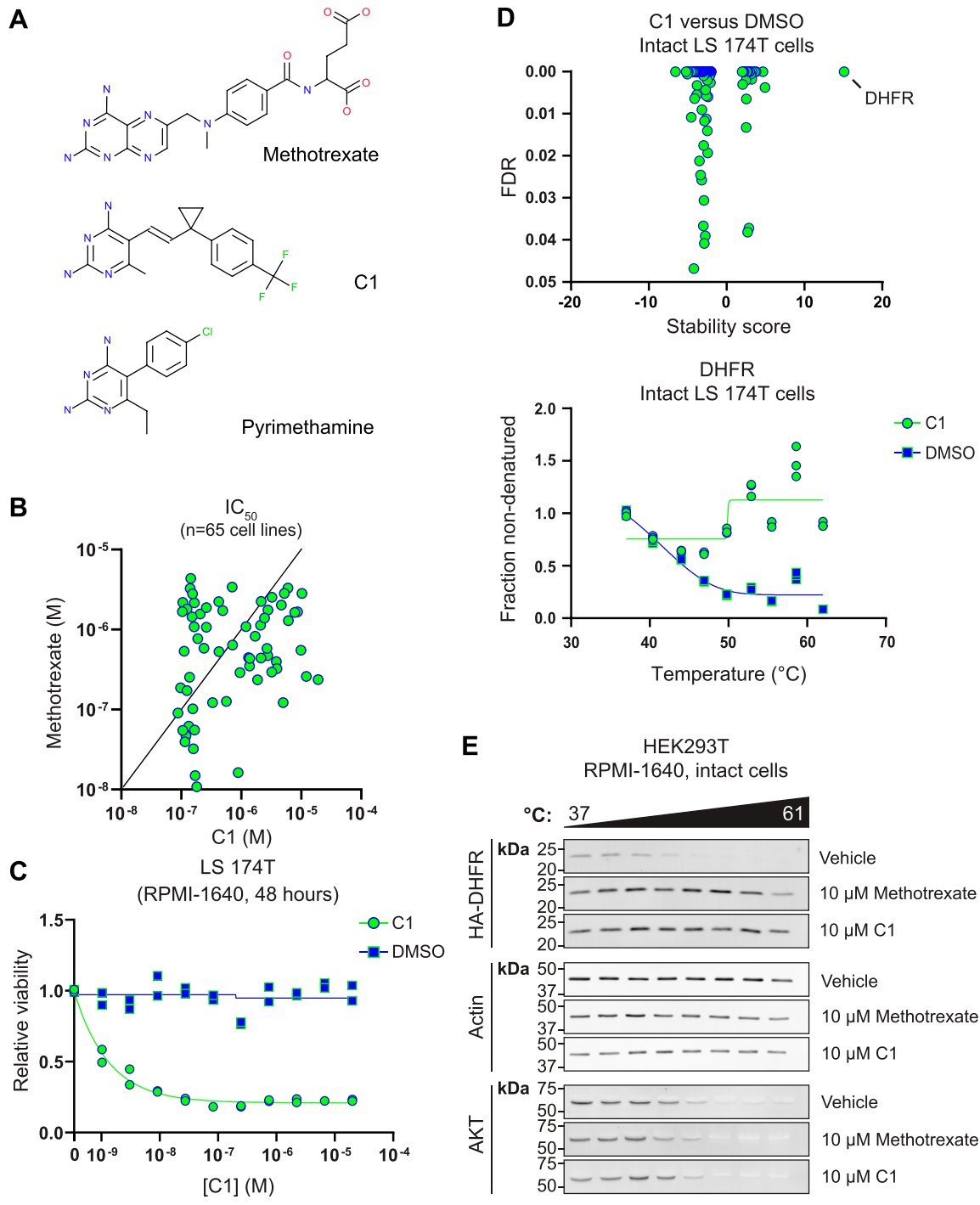

**Figure 1. Identification of compound C1 as a novel DHFR ligand.**
**(A)** Structures of methotrexate, pyrimethamine, and C1, a novel 2,4-diaminopyrimidine antifolate. **(B)** Comparison of C1 and methotrexate $IC_{50}$ values for a panel of cancer cell lines. Methotrexate $IC_{50}$ values were retrieved from the Genomics of Drug Sensitivity in Cancer database and determined using a Syto60 viability assay after 72 h treatment. C1 $IC_{50}$ values were determined using a sulforhodamine B viability assay after 72 h treatment. **(C)** CellTiter-Glo viability assay on C1- or vehicle-treated LS 174T cells after 48 h treatment. Data were collected for n = 2 biological replicates. **(D)** Top panel: thermal proteome profile of intact LS 174T cells treated with 10 μM C1 for 1 h, including stability score and false discovery rate of all hits and candidates. Negative stability scores represent proteins that are destabilized by C1 treatment, and positive stability scores represent proteins that are stabilized by C1 treatment. Bottom panel: melting curve for DHFR in 10 μM C1- or DMSO-treated intact LS 174T cells. **(E)** Western blot analysis of a representative cellular thermal shift assay on intact HEK293T cells overexpressing HA–DHFR treated with 10 μM methotrexate or C1 for 1 h.

### C1 inhibits DHFR and downstream events in folate-mediated one-carbon metabolism

To analyze pharmacological inhibition of DHFR, we measured in vitro activity of recombinant hDHFR in the presence of various concentrations C1 or methotrexate, using excess concentrations of DHF and NADPH substrate. Methotrexate inhibited hDHFR activity close to 100% at all tested concentrations, in line with the previously reported 6 pM $K_i$ of methotrexate towards mouse DHFR at 100 nM DHF (Fig 2A) (Piper et al, 1985). C1 also inhibited in vitro hDHFR activity, although with a higher $IC_{50}$ of ~1 $\mu$M in the presence of 50 $\mu$M DHF, suggesting that C1 binds to hDHFR with a lower affinity than methotrexate (Fig 2A). Next, we applied isothermal titration calorimetry (ITC) to determine the dissociation constants ($K_d$) of hDHFR to C1 and methotrexate (Table 1 and Fig S2A). On average, the $K_d$ of hDHFR for methotrexate was 5 nM, whereas the $K_d$ of hDHFR for C1 was 28 nM, indicating that the affinity of C1 for hDHFR is weaker than methotrexate. Addition of 125 $\mu$M NADPH did not affect binding of both C1 and methotrexate (Fig S2B). These results show that, like methotrexate, C1 directly inhibits DHFR, albeit with lower affinity.

Next to monitoring drug–protein interactions, TPP may be used to assess drug-induced alterations in metabolite-protein interactions within cells (Huber et al, 2015). We hypothesized that inhibition of DHFR would indirectly lead to thermal shifts of enzymes catalyzing THF-dependent reactions as a result of altered substrate availability. To obtain insight into the metabolic consequences of C1 treatment, we used PANTHER statistical overrepresentation analysis to identify biological processes and pathways linked to alterations in proteome stability of C1-treated LS 174T cells (Mi et al, 2019). Proteins involved in various anabolic processes were significantly overrepresented in the thermal proteome profile of C1-treated cells, including pathways linked to nucleotide biosynthesis, transcription, and translation (Table 2). We also observed C1-induced destabilization of C1-tetrahydrofolate synthase (MTHFD1) (Fig 2B), which suggests loss of interactions, indicative of decreased substrate availability and thus corroborates C1-mediated inhibition of cytosolic 5,10-methylene–THF and 10-formyl–THF production. To compare the TPP of C1-treated LS 174T cells with methotrexate, we analyzed the published TPP of K562 cells treated with 10 $\mu$M methotrexate for 3 h and performed Z-score transformation on both datasets to enable comparison (Huber et al, 2015). Both C1 and methotrexate significantly stabilized DHFR and TYMS, whereas C1, but not methotrexate, did cause significant destabilization of MTHFD1 (Fig 2B).

To compare the cellular effects of C1 and methotrexate in more detail, we treated A549 cells with a titration of both drugs for 48 h, including a rescue with 25 $\mu$M 5-formyl–THF, a metabolite that can be converted to 5,10-methylene–THF and 10-formyl–THF, independently of DHFR function (Fig 2C) (Ducker & Rabinowitz, 2017). Using an ATP-based cell viability assay, A549 cells displayed high sensitivity to C1 with a sub-nM $IC_{50}$, whereas the $IC_{50}$ for methotrexate was around 50 nM. The $IC_{50}$ of both drugs increased ~400-fold upon 5-formyl–THF supplementation (Fig 2D), suggesting that both drugs impair one-carbon transfer via the folate cycle, presumably through inhibition of DHFR. To assess whether C1 and methotrexate inhibit specific reactions of the one-carbon cycle, we treated A549 and HCT-116 cells with 100 nM drug and investigated

rescue effects by supplementation with the one-carbon cycle metabolites thymidine, hypoxanthine, formate, and glycine (Fig 2C). C1- and methotrexate-treated A549 and HCT-116 cells were rescued by a combination of thymidine and hypoxanthine (Figs 2E and S3A), suggesting that both drug treatments deplete intracellular 5,10-methylene–THF and 10-formyl–THF pools, required for purine and thymidine synthesis (Fig 2C) (Lawrence et al, 2014; Yang & Vousden, 2016; Ducker & Rabinowitz, 2017; Zheng et al, 2018). Supplementation with formate or glycine, which can be used by MTHFD1 to generate cytosolic 10-formyl–THF or by the glycine cleavage system to generate mitochondrial 5,10-methylene–THF, respectively, did not rescue viability of C1- and methotrexate-treated cells (Fig 2E). These findings suggest that depletion of intracellular THF limits these reactions (Fig 2C) (Yang & Vousden, 2016; Ducker & Rabinowitz, 2017). The nutrient sensing branch of the mechanistic target of rapamycin complex 1 (mTORC1) signaling network responds to intracellular purine levels, leading to mTORC1 inhibition upon antifolate treatment (Emmanuel et al, 2017; Hoxhaj et al, 2017). To compare the effects of C1 and methotrexate on purine sensing by mTORC1, we treated A549 cells overnight with both drugs, followed by a 2.5 h supplementation of adenosine and guanosine to restore intracellular purine levels. Western blot analysis of the mTORC1 substrate 4E-BP1 revealed mTORC1 inhibition by C1 and methotrexate, which could be partially restored by adenosine but not guanosine supplementation (Fig S3B). This observation is in accordance with the report that mTORC1 is inhibited by short-term depletion of adenylates and suggests that methotrexate and C1 have similar effects on intracellular purines and associated mTORC1 activity (Hoxhaj et al, 2017).

Notwithstanding these in vitro results, we observed that a subset of cancer cell lines (including A549) display over 50-fold higher sensitivity to C1 than methotrexate, pyrimethamine, and trimethoprim (Fig 2F). These observations suggest that inhibition of DHFR activity in vitro does not necessarily reflect the intracellular situation within live cells, where folate concentrations are expected to be lower (Bailey et al, 2015). Combined, our results show that C1 is a novel DHFR inhibitor that interferes with THF-mediated one-carbon transfer reactions required for de novo nucleotide synthesis, DNA replication, transcription, and translation. Furthermore, our results show that C1 is less potent in inhibiting DHFR than methotrexate, whereas some cell lines display a 50-fold higher sensitivity to C1, emphasizing the pivotal role of intracellular context on evaluating cellular drug responses.

### C1 and methotrexate inhibit DHFR by a partially overlapping binding mode

Our in vitro results might suggest that C1 and methotrexate may have different DHFR binding modes. To compare DHFR binding by C1 with methotrexate and natural folates, we generated docking models of the hDHFR–NADPH–C1 complex by using published crystal structures of hDHFR in complex with 2,4-diaminopyrimidine compounds, and we compared our models with structures of hDHFR in complex with natural folates or methotrexate (Table 3). The hDHFR active site cleft transitions from an open (Fig 3A) to a closed conformation upon binding to ligand (Fig 3B and C) or to methotrexate (Fig 3D) (Cody et al, 2005; Bhabha et al, 2013). We

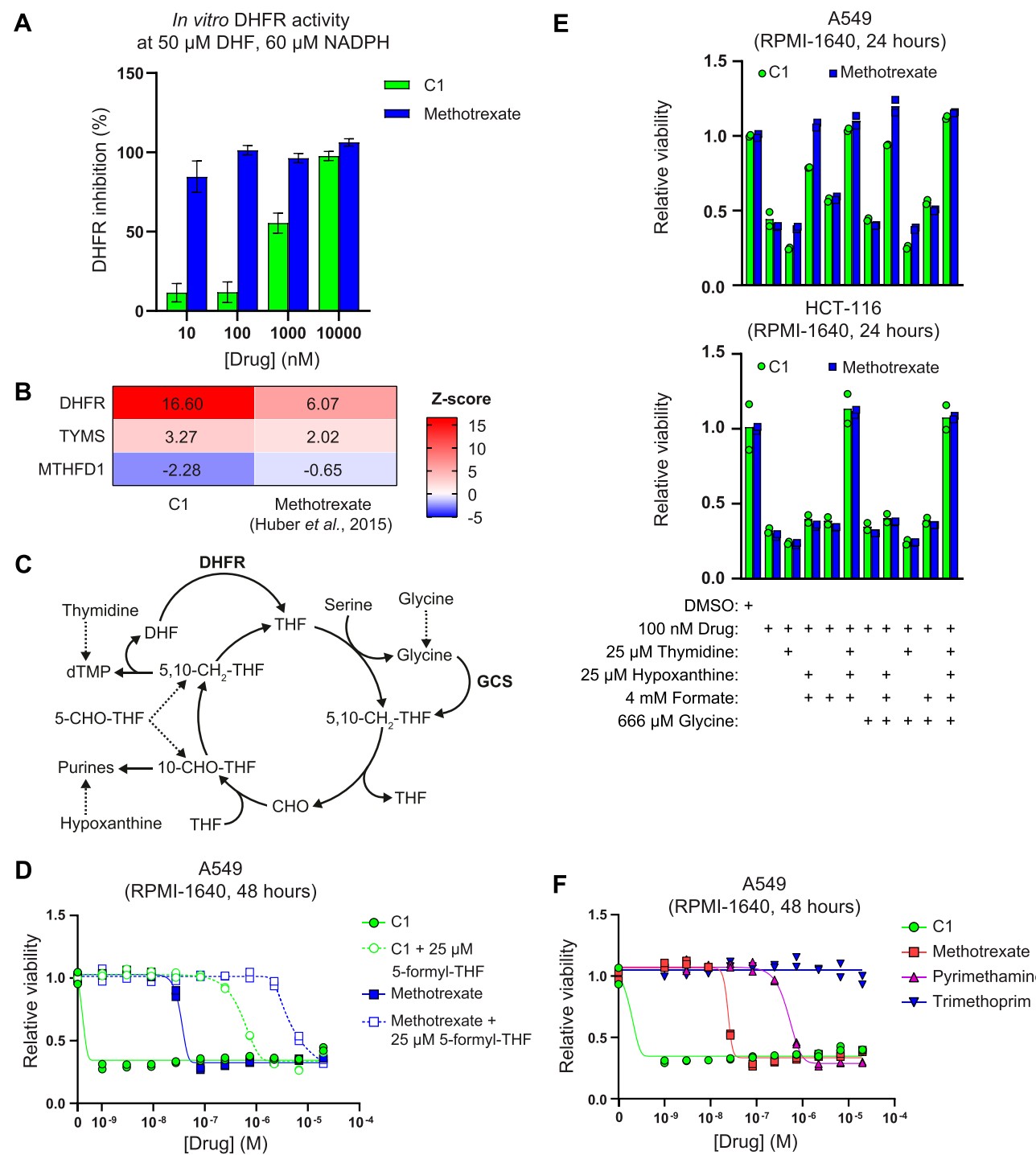

**Figure 2. C1 and methotrexate inhibit DHFR and folate-mediated one-carbon metabolism.**
**(A)** In vitro inhibition of hDHFR activity by C1 and methotrexate, at 50 $\mu$M dihydrofolate and 60 $\mu$M NADPH. Data are shown as mean ± s.d. and were collected in n = 2 technical replicates. Results are representative of three independent experiments. **(B)** Thermal proteome profiling analysis of one-carbon metabolism–related proteins in cells treated with C1 or methotrexate, including Z-score transformed stability scores. Positive Z-scores represent stabilized proteins, negative Z-scores represent destabilized proteins. The thermal proteome profile for C1 was determined at 10 $\mu$M in LS 174T cells (Fig 1D). The thermal proteome profile for methotrexate was retrieved from Huber et al (2015) and determined at 10 $\mu$M in K562 cells. **(C)** Simplified schematic representation of tetrahydrofolate (THF)-mediated one-carbon transfer reactions involved in purine and thymidine synthesis. Rescue interventions are indicated with dashed lines. Abbreviations: 5,10-methylene-tetrahydrofolate (5,10-CH$_2$-THF), formate (CHO), 10-formyl-tetrahydrofolate (10-CHO-THF), 5-formyl-tetrahydrofolate (5-CHO-THF), glycine cleavage system (GCS). **(D)** CellTiter-Glo viability analysis of C1- or methotrexate-treated A549 cells after 48 h treatment, in the absence or presence of 25 $\mu$M 5-formyl-tetrahydrofolate. Data were collected in n = 2 biological replicates. **(E)** CellTiter-Glo viability analysis of C1- or methotrexate-treated A549 or HCT-116 cells after 24 h treatment, including rescue treatments with one-carbon cycle metabolites. Data were collected in n = 2 biological replicates. **(F)** CellTiter-Glo viability analysis of C1-, methotrexate-, pyrimethamine-, or trimethoprim-treated A549 cells after 48 h treatment. Data were collected in n = 2 biological replicates.

**Table 1.   Dissociation constants (K_d) of hDHFR to C1 and methotrexate in absence of NADPH, determined by isothermal titration calorimetry.**

| Experiment | Average $K_d$ (±s.d.) (nM) | Measured $K_d$ range (nM) | Replicates |
|---|---|---|---|
| hDHFR to C1 | 28.10 ± 18.38 | 15.1–41.1 | 2 |
| hDHFR to methotrexate | 4.75 ± 0.80 | 4.18–5.32 | 2 |

**Table 2.   Thermal proteome profiling reveals various anabolic processes and pathways affected by C1 treatment in LS 174T cells.**

| PANTHER GO-Slim Biological Process | Fold enrichment | FDR |
|---|---|---|
| Positive regulation of RNA polymerase II transcription preinitiation complex assembly (GO:0045899) | >100 | $5.71 \times 10^{-8}$ |
| Pyrimidine nucleotide biosynthetic process (GO:0006221) | 52.81 | $9.47 \times 10^{-4}$ |
| Proteasome assembly (GO:0043248) | 45.26 | $2.03 \times 10^{-2}$ |
| Ribosomal large subunit assembly (GO:0000027) | 41.33 | $9.34 \times 10^{-7}$ |
| RNA polymerase II preinitiation complex assembly (GO:0051123) | 34.44 | $2.54 \times 10^{-5}$ |
| Nucleoside monophosphate metabolic process (GO:0009123) | 25.01 | $6.53 \times 10^{-3}$ |
| Ribosomal small subunit biogenesis (GO:0042274) | 17.6 | $6.96 \times 10^{-5}$ |
| Ribonucleoside triphosphate biosynthetic process (GO:0009201) | 12.43 | $8.08 \times 10^{-3}$ |
| Cytoplasmic translation (GO:0002181) | 11.74 | $9.48 \times 10^{-3}$ |
| Purine ribonucleotide biosynthetic process (GO:0009152) | 10.93 | $7.38 \times 10^{-4}$ |
| DNA-dependent DNA replication (GO:0006261) | 10.56 | $1.36 \times 10^{-2}$ |
| Translational elongation (GO:0006414) | 8.73 | $5.67 \times 10^{-7}$ |
| Proteasome-mediated ubiquitin-dependent protein catabolic process (GO:0043161) | 6.53 | $1.13 \times 10^{-3}$ |
| Oxoacid metabolic process (GO:0043436) | 3.99 | $1.08 \times 10^{-2}$ |
| **PANTHER pathways** | **Fold enrichment** | **FDR** |
| Sulfate assimilation (P02778) | >100 | $7.34 \times 10^{-3}$ |
| Tetrahydrofolate biosynthesis (P02742) | 63.37 | $1.62 \times 10^{-2}$ |
| ATP synthesis (P02721) | 52.81 | $1.93 \times 10^{-2}$ |
| Formyltetrahydroformate biosynthesis (P02743) | 45.26 | $2.25 \times 10^{-2}$ |
| Cell cycle (P00013) | 43.21 | $4.40 \times 10^{-7}$ |
| Ubiquitin proteasome pathway (P00060) | 31.17 | $1.59 \times 10^{-12}$ |
| DNA replication (P00017) | 15.33 | $2.21 \times 10^{-2}$ |

PANTHER GO-slim Biological Process and PANTHER pathways statistical overrepresentation test on (de)stabilized hits and candidates listed in Fig 1D. The table includes fold enrichment and false discovery rate (FDR).

observed a similar closed conformation in the docking model of the hDHFR–NADPH–C1 complex, although both open (Fig 3E) and closed (Fig 3F) conformations of Phe31 were predicted energetically favorable. hDHFR is known to bind folate and 5,10-dideaza–THF through formation of hydrogen bonds with Glu30, Asn64, and Arg70 and hydrophobic interactions with Phe31, which causes the active site to adopt a closed conformation (Fig 3G and H) (Bhabha et al, 2013). These contacts are conserved in the hDHFR–NADPH–methotrexate complex. In addition, methotrexate forms a hydrogen bond with Tyr121 (Fig 3I and J). Our predicted hDHFR–NADPH–C1 complex indicates that hydrogen bonds with Tyr121 and Glu30, and hydrophobic contacts with Phe31, are

conserved (Fig 3K). By contrast, Asn64 and Arg70 do not participate in complex formation of hDHFR with C1 (Fig 3L), providing a potential explanation for the higher IC_50 value of C1 compared with methotrexate (Figs 2A and 3H and J). Taken together, our results suggest that the binding mode of C1 to hDHFR partially overlaps with methotrexate, although displaying an overall reduced binding surface. These results explain the lower affinity of C1 compared with methotrexate observed in our in vitro experiments. Of note, our simulation system was not biased with any specific restraints that could recreate the hydrogen bond network, that is, observed with folates and methotrexate, indicating a similar binding mode for C1.

**Table 3.  Crystal structures used for generation of docking models of the hDHFR–NADPH–C1 complex and comparison with methotrexate and natural folates.**

| PDB accession | Ligand 1 | Ligand 2 | Resolution (Å) | Reference | Docked ligand | Fig 3 |
|---|---|---|---|---|---|---|
| 4M6J | NADPH | N.A. | 1.20 | Bhabha et al (2013) | N.A. | A, G, H |
| 4M6K | NADP+ | Folate | 1.49 | Bhabha et al (2013) | N.A. | B, G, H |
| 4M6L | NADP+ | 5,10-Dideaza–THF | 1.70 | Bhabha et al (2013) | N.A. | C, G, H |
| 1U72 | NADPH | Methotrexate | 1.90 | Cody et al (2005) | N.A. | D, I, J |
| 4KBN | NADPH | 5-{3-[3-(3,5-pyrimidine)]-phenyl-prop-1-yn-1-yl}-6-ethylpyrimidine-2,4-diamine | 1.84 | Lamb et al (2013) | C1 | E, K, L |
| 5HPB | NADPH | 5-methyl-6-(phenylthio-4'trifluoromethyl)thieno[2,3-d] pyrimidine-2,4-diamine | 1.65 | Unpublished | C1 | F, K, L |

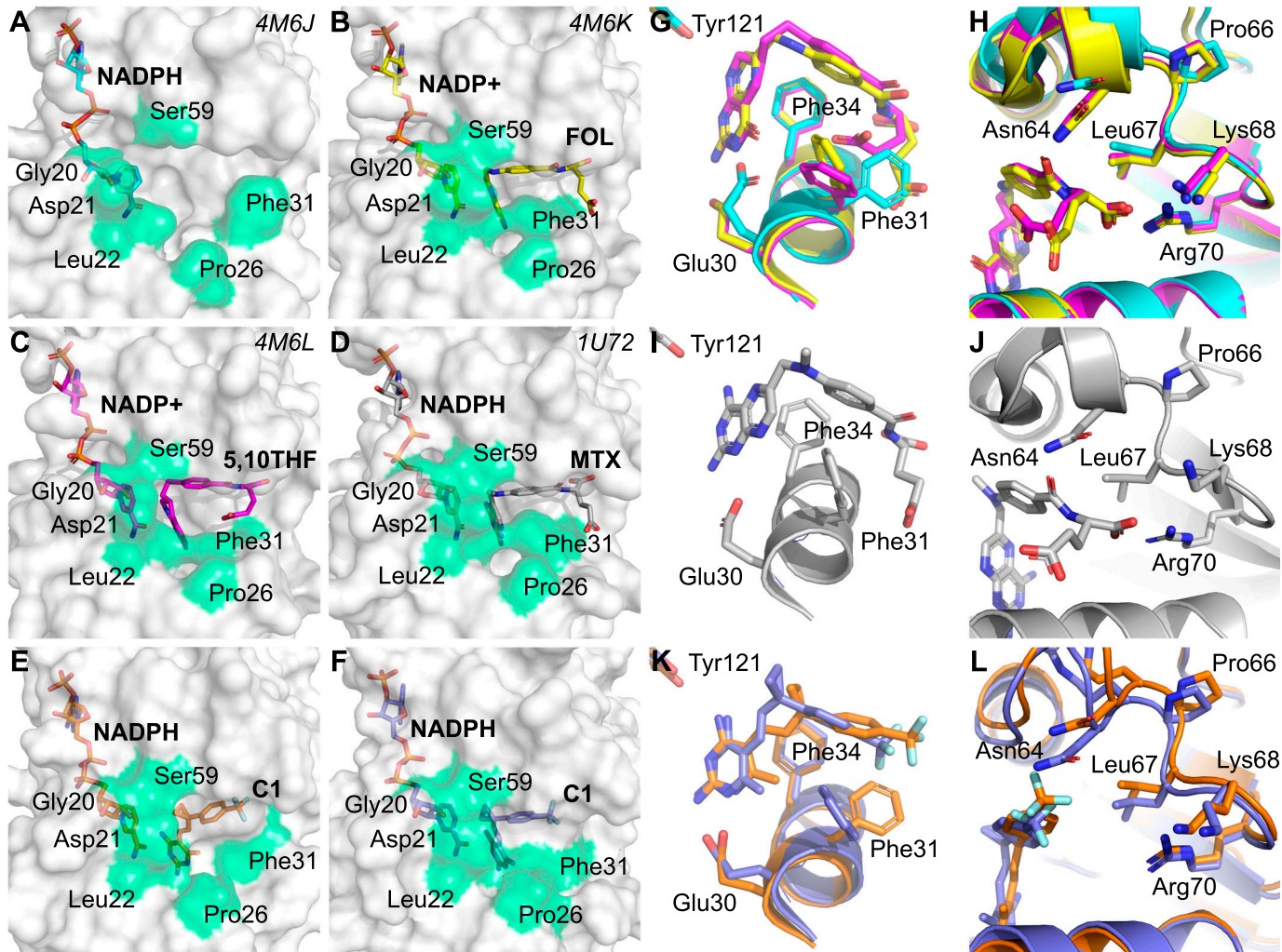

**Figure 3.   DHFR-binding modes of C1 and methotrexate are predicted to be partially overlapping.**
**(A, B, C, D, E, F)** Surface representation of (A) crystal structure of hDHFR in complex with NADPH, (B) crystal structure of hDHFR in complex with NADP+ and folic acid (FOL), (C) crystal structure of hDHFR in complex NADP+ and 5,10-dideaza-THF (5,10THF), (D) crystal structure of hDHFR in complex with NADPH and methotrexate (MTX), (E) docking pose of C1 on hDHFR (PDB accession 4M6J), and (F) docking pose of C1 on hDHFR (PDB accession 5HPB). Residues involved in opening and closing of the active site are highlighted in green and were adopted from Bhabha et al (2013). **(G, H, I, J)** Active site crystal structures of the (G, H) hDHFR–NADPH (cyan), hDHFR–NADPH–folic acid (yellow), hDHFR–NADP+–5,10-dideaza–THF (magenta), and (I, J) hDHFR–NADPH–methotrexate complexes. **(K, L)** Docking poses of the hDHFR–NADPH–C1 complexes (orange is PDB accession 4KBN, purple is PDB accession 5HPB).

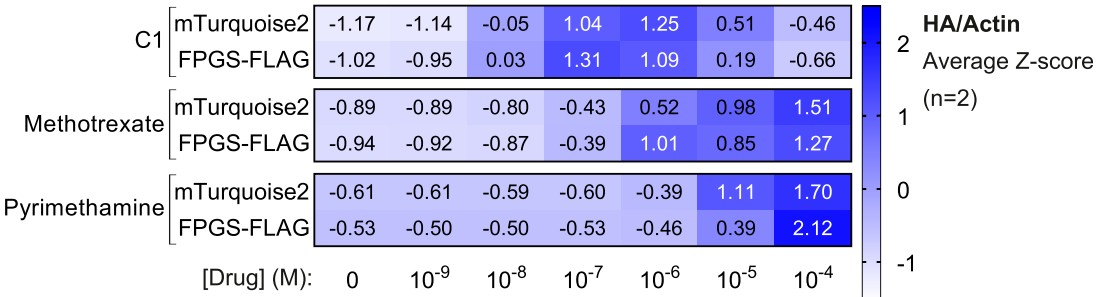

**Figure 4. C1 potently forms a complex with DHFR in cells.**
Drug-induced HA–DHFR stabilization in cellular thermal shift assays on intact HEK293T cells overexpressing HA–DHFR and mTurquoise2 (negative control) or FPGS–FLAG. Cells were incubated with a drug concentration range for 4 h followed by a cellular thermal shift assay at 52°C. Data are average Z-scores and were collected in n = 2 biological replicates. Original Western blots are shown in Fig S6C.

## DHFR engagement by antifolates in cells is determined by physicochemical properties and cellular context

Our results reveal that C1 has a higher affinity for DHFR than DHF, but a lower affinity than methotrexate. Nevertheless, a subset of cell lines displays up to 50-fold higher sensitivity to C1, emphasizing that cellular context is a major determinant of antifolate sensitivity. A key regulatory element, that is, missing in in vitro assays is the presence of membrane-encapsulated subcellular compartments. Methotrexate and natural folates do not diffuse across membranes because of their charged glutamate residue(s) but are actively imported by the FOLR pathway, SLC19A1, and SLC46A1, whereas intracellular accumulation is promoted by FPGS-mediated polyglutamylation (Zheng & Cantley, 2019). Conversely, polyglutamates can be hydrolyzed to monoglutamates by the lysosomal or secreted enzyme γ-glutamyl-hydrolase (GGH), which is thought to facilitate membrane transport of methotrexate and natural folates (Zheng & Cantley, 2019). These properties categorize methotrexate as a classical antifolate (Gangjee et al, 2007, 2008). By contrast, lipophilic compounds such as trimetrexate and pyrimethamine cross membranes by passive diffusion and do not depend on FPGS activity for cellular retention, classifying them as non-classical antifolates. C1 lacks a glutamate group compatible with FPGS-mediated polyglutamylation and has a positive octanol–water distribution coefficient (LogD), illustrating that C1 is lipophilic and therefore classifies as a non-classical antifolate (Fig S4).

In line with suggestions in the literature, we hypothesized that FPGS-dependent concentrations of intracellular (anti)folate determine sensitivity of cells to classical or non-classical antifolates like C1 (Fabre et al, 1984; Rots et al, 1999; Liani et al, 2003; Zhao & Goldman, 2003; Fotoohi et al, 2009; Stark et al, 2009; Wojtuszkiewicz et al, 2016; Li et al, 2020; Yu et al, 2020; Zarou et al, 2021). To address this issue, we used data from the Cancer Cell Line Encyclopedia (Ghandi et al, 2019), to calculate correlation coefficients between polyglutamylation-associated gene expression and sensitivity (IC$_{50}$) to C1 or methotrexate, for a panel of cell lines (Fig S5A). In addition, we used two larger datasets to investigate how polyglutamylation-associated gene expression correlates with cellular sensitivity for pyrimethamine and methotrexate (Fig S5B). Notably, *FPGS* expression positively correlated with sensitivity to methotrexate (Fig

S5B), although displaying an inverse correlation with C1 sensitivity (Fig S5A). This analysis did not reveal a relationship between sensitivity to C1 and expression of *GGH* (Fig S5A), although *GGH* expression correlated with resistance to methotrexate (Fig S5B). These findings thus further corroborate FPGS deficiency as a mechanism of methotrexate resistance and suggest that FPGS determines whether cells display sensitivity to C1 or methotrexate. Interestingly, this methotrexate therapy-induced escape route of cancer cells thus may uncover a cellular vulnerability to C1.

To analyze the effect of FPGS overexpression on intracellular DHFR binding by classical and non-classical antifolates, we applied CETSA in a semi-quantitative experimental setup for measuring intracellular DHFR-antifolate binding. We used intact HEK293T cells overexpressing DHFR with and without FPGS and tested a drug dilution series, while keeping a fixed, denaturing temperature (Fig 4). Of note, HEK293T cells express normal levels of FPGS protein and are thus considered FPGS-competent (Li et al, 2021; Fig S6A). Western blot analysis confirmed overexpression of HA–DHFR and FPGS–FLAG in mock-treated samples used for CETSA (Fig S6B). CETSA revealed that C1 potently stabilized DHFR at 10 nM and higher concentrations, in line with the 28 nM K$_d$ of the C1–DHFR complex determined by ITC (Figs 4 and S6C and Table 1). Methotrexate and pyrimethamine also stabilized DHFR, although at higher concentrations than C1, indicative of decreased intracellular complex formation with DHFR (Figs 4 and S6C). In contrast with the 5 nM K$_d$ of the methotrexate–DHFR complex determined by ITC (Table 1), 10 nM of methotrexate does not lead to intracellular DHFR stabilization (Figs 4 and S6C and Table 1), supporting the view that the presence of membrane-encapsulated intracellular compartments limits the action of this hydrophilic molecule. Overexpression of FPGS did not significantly alter DHFR stabilization by methotrexate, pyrimethamine, and C1 at all tested concentrations (Figs 4 and S6C). This result shows that, in FPGS-competent HEK293T cells, overexpression of FPGS does not alter intracellular complex formation of antifolates with DHFR. Furthermore, this finding shows that, in contrast to methotrexate, C1 is not limited in reaching and binding intracellular DHFR. We conclude that C1, despite its lower affinity, may gain a functional advantage over hydrophilic methotrexate in FPGS-deficient contexts, which will further limit the intracellular retention of methotrexate, but not C1.

## C1 selectively suppresses growth of FPGS-deficient cells

Next to mediating methotrexate resistance, loss of FPGS is a known mechanism of resistance towards 5-FU treatment in CRC cells (Sohn et al, 2004), which may uncover a vulnerability for C1 treatment. To address this issue, we used patient-derived CRC organoids characterized by FPGS deficiency to analyze the relationship between polyglutamylation and therapeutical efficiency of antifolates and 5-FU (Van de Wetering et al, 2015). Based on RNAseq analysis, patient 6–derived tumor organoids display severe transcriptional down-regulation of *FPGS* expression (Van de Wetering et al, 2015; Fig S7A) and RT-qPCR analysis of patient 6–derived tumor organoids (P6T) confirms that this line displays transcriptional down-regulation of *FPGS* expression compared with patient 26-derived normal colon organoids (P26N; Fig S7B). At the same time, the expression of *GGH*, which catalyzes hydrolysis of folate polyglutamates, is increased in P6T organoids (Fig S7B). These findings indicate that P6T organoids have a decreased capacity to polyglutamylate FPGS-dependent antifolates, which is expected to decrease the intracellular concentration and efficacy of these drugs. Indeed, a viability assay of P6T organoids at 7 d after start of treatment revealed that this organoid line is ~threefold more sensitive to C1 than to methotrexate, in line with our model that FPGS deficiency creates a vulnerability to C1 (Fig S8A and B).

To further test how efficacy of different antifolate classes depends on cellular polyglutamylation capacity, we used transposase-mediated integration of doxycycline-inducible FPGS, GGH, and empty vector (EV) overexpression constructs, thereby including independent expression cassettes for fluorescent mNeonGreen, mCherry, or NLS–TagBFP reporters, respectively. Immunofluorescence and Western blot analyses confirmed that FPGS and GGH were overexpressed when organoids were cultured in the presence of doxycycline (Fig S9A and B). P6T organoids overexpressing FPGS, GGH, and EV were cultured in the presence of 100 and 500 nM C1 or 200 and 500 nM methotrexate, and outgrowth was analyzed on day 8 by imaging fluorescent organoids, followed by a viability assay. The lower methotrexate concentration was set at 200 nM to compensate for the decreased sensitivity of P6T organoids for methotrexate compared with C1 (Fig S8A and B). P6T organoids overexpressing EV showed a dose-dependent decrease in outgrowth upon treatment with either C1 or methotrexate (Fig 5A), which was even more apparent in the viability assay (Fig 5B). An increased sensitivity of the chemical viability assay may be explained by the fact that alterations in intracellular metabolite concentrations (fast) may occur before outgrowth (slow) is impaired. Organoids overexpressing FPGS showed a comparable dose-dependent decrease in outgrowth during C1 treatment as observed for non-FPGS–overexpressing cells but were strongly sensitized to methotrexate treatment, which decreased organoid outgrowth by 80% (Fig 5A). Viability analysis further revealed that FPGS overexpression in organoids mediated a rescue from treatment with 100 nM C1 ($P <0.0001$), but not 500 nM C1 (Fig 5B). These findings are in line with our model that, when re-expressed in FPGS-deficient cells, FPGS prevents DHFR–C1 complex formation by enhancing the intracellular concentrations of folate, which can be outcompeted by increasing doses of C1. P6T organoids overexpressing GGH showed a dose-dependent decrease in outgrowth

upon both drug treatments (Fig 5A). In contrast to FPGS, overexpression of GGH created a minor rescue from treatment with 100 nM C1 ($P = 0.0348$), but not 500 nM C1 (Fig 5B). For 200 nM methotrexate, GGH overexpression caused a more significant rescue ($P <0.0001$), which was not observed for 500 nM methotrexate (Fig 5B). Combined, these results reveal that FPGS-deficient P6T organoids display sensitivity to C1 although being more resistant to methotrexate, which is reverted upon re-introduction of FPGS, indicating that FPGS deficiency sensitizes to FPGS-independent antifolates like C1. The role of GGH appears to be more complex because it decreases toxicity of both methotrexate and C1, however, and may involve differential effects on the import of natural folates and drug compounds.

Next, we examined if clinically approved polyglutamylation-independent antifolates also preferentially target FPGS-deficient cells, like C1. To address this point, we performed viability assays using FPGS-deficient P6T and FPGS-competent P26T CRC organoids that display similar FPGS expression levels to normal colon organoids (Van de Wetering et al, 2015; Fig S7A and B). We assessed the response to C1, methotrexate, the polyglutamylation-independent methotrexate-derivative trimetrexate, and the TYMS inhibitor 5-FU. Compared with P26T organoids, P6T organoids were relatively insensitive to methotrexate, trimetrexate and 5-FU, but displayed sensitivity to C1. Overall, P6T organoids were threefold more sensitive to C1 than to methotrexate and trimetrexate (Fig S8A and B). This suggests that C1 classifies as a more potent polyglutamylation-independent antifolate than trimetrexate for the treatment of FPGS-deficient cells, although we cannot rule out that trimetrexate activity and/or resistance may occur through alternative mechanisms.

## C1 overcomes methotrexate resistance of FPGS-deficient tumor organoids

In ALL, primary tumors were reported to contain persister clones that survive initial chemotherapy treatment and grow out at later timepoints to cause therapy resistance and relapse (Li et al, 2020; Yu et al, 2020). In methotrexate-treated patients, FPGS deficiency may develop through inactivating mutations or transcriptional down-regulation, resulting from copy number alterations or promoter deletions (Li et al, 2020; Yu et al, 2020). We hypothesized that C1 treatment may prevent outgrowth of FPGS-deficient clones in a heterogeneous tumor and thereby overcome methotrexate resistance. To mimic tumor heterogeneity, we co-cultured FPGS-deficient P6T organoid lines overexpressing EV (control), FPGS, or GGH and evaluated their relative outgrowth during treatment with various concentrations of C1 or methotrexate. Outgrowth was measured on day 8 by imaging fluorescent reporters and quantification of organoid surface area (Fig 5C and D). Although 100 nM C1 suppressed growth of FPGS-deficient organoids (overexpressing EV or GGH), FPGS overexpression caused full resistance to 100 nM C1 under co-culture conditions ($P = 0.0017$) (Fig 5C and D). FPGS-mediated resistance was lost at 500 nM C1, in line with our previous results (Fig 5A and B). Conversely, 200 and 500 nM methotrexate suppressed growth of FPGS-overexpressing organoids ($P <0.0001$ and $P = 0.0028$, respectively), whereas FPGS-deficient organoids were partially resistant (Fig 5C and D). P6T organoids overexpressing GGH responded identically to all tested

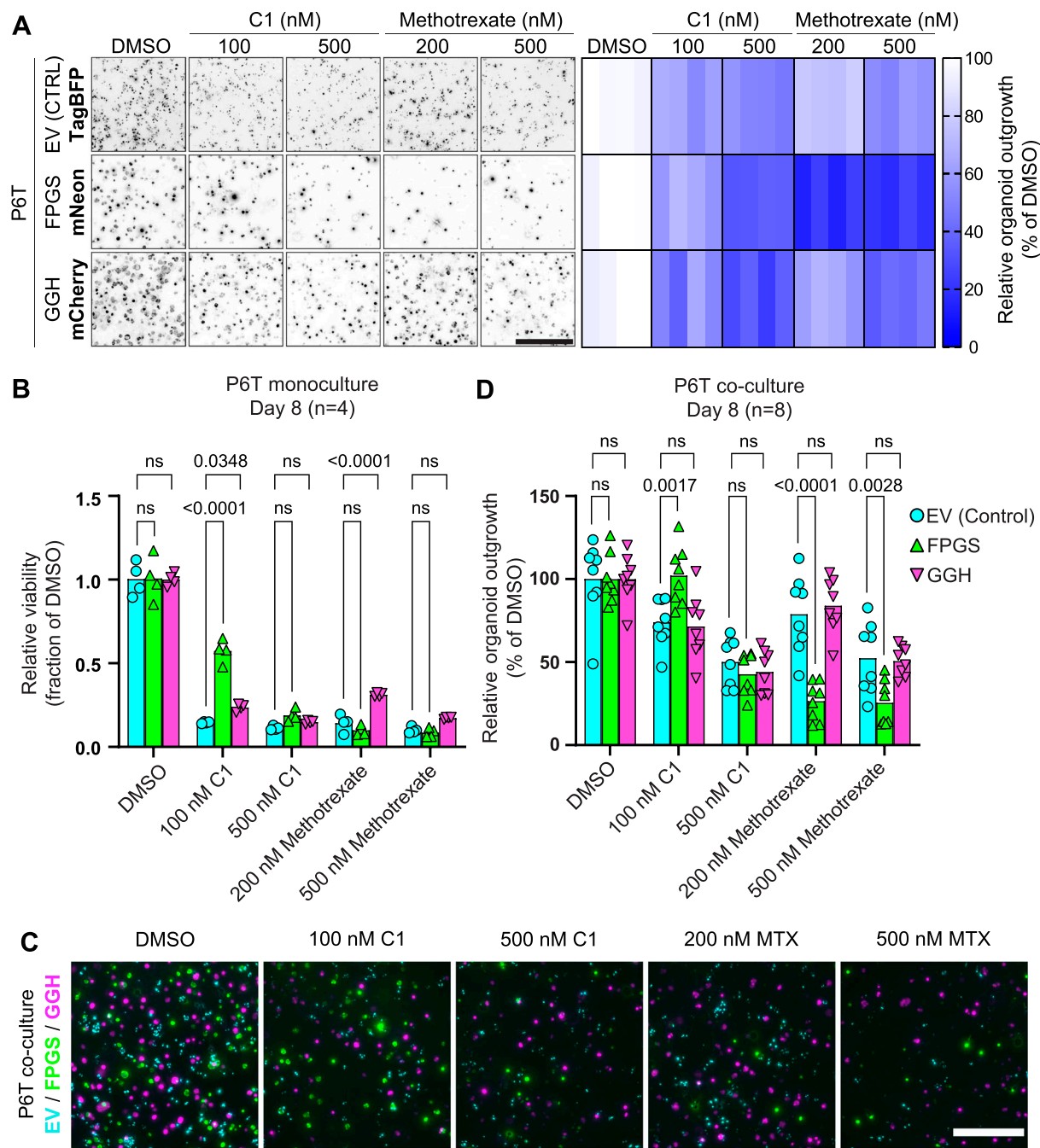

**Figure 5. FPGS deficiency causes methotrexate resistance, but creates a vulnerability to C1.**
**(A)** Representative widefield fluorescence images of P6T organoids overexpressing doxycycline-inducible constructs with a fluorescent reporter after 8 d of drug treatment. Overexpression of empty vector IRES–TagBFP (EV), FPGS–FLAG IRES–mNeonGreen (FPGS), or GGH–FLAG IRES–mCherry (GGH) was combined with various drug treatments for 8 d. Organoids were treated with DMSO (vehicle control), 100 or 500 nM C1 and 200 or 500 nM methotrexate. Relative organoid outgrowth is measured as % surface area of DMSO. Images are Z-stack projections of deconvoluted widefield images. Scalebar represents 500 μm. Data were collected in n = 2 independent experiments with biological duplicates. **(B)** CellTiter-Glo viability analysis of P6T organoids overexpressing empty vector, FPGS, or GGH after 8 d of drug treatment. Data were collected in n = 2 independent experiments with biological duplicates. Viability of FPGS- and GGH-overexpressing organoids was compared with empty vector–overexpressing organoids using two-way ANOVA with Dunnett's multiple comparisons test. **(C)** Representative widefield fluorescence images of co-cultured P6T organoids overexpressing EV (cyan), FPGS (green), or GGH (magenta) after 8 d of C1 or methotrexate (MTX) treatment. Overexpression was combined with various drug treatments for 8 d. Images are Z-stack projections of deconvoluted widefield images. Scalebar represents 500 μm. Data were collected in n = 2 independent experiments with biological quadruplicates. **(D)** Quantification of fluorescence surface area in widefield images of co-cultured P6T organoids overexpressing EV (cyan), FPGS (green), or GGH (magenta) after 8 d of drug treatment. Analysis was performed on Z-stack projections of deconvoluted widefield images. Relative surface area of FPGS- and GGH-overexpressing organoids was compared with empty vector-overexpressing organoids using two-way ANOVA with Dunnett's multiple comparisons test. Data were collected in n = 2 independent experiments with biological quadruplicates.

antifolate treatments as organoids overexpressing EV, which suggests that the relationship between antifolate efficacy and polyglutamylation is primarily shaped by FPGS, not GGH. Taken together, our results show that FPGS deficiency creates vulnerability for C1, whereas cells with functional FPGS are sensitive to methotrexate. Our findings indicate that a regimen in which methotrexate-based chemotherapy is alternated with C1 treatment will suppress outgrowth of FPGS-deficient, methotrexate-resistant persister clones that may cause disease relapse at later stages.

## Discussion

Methotrexate has been successfully used for chemotherapy, but its therapeutic index is limited by off-target toxicity, including bone marrow suppression, damage to gastrointestinal epithelia and hepatotoxicity (Takami et al, 1995; Wilson et al, 2014). These problems fostered efforts to discover new antifolates with improved therapeutic index, illustrated by seven clinically approved and 15 DHFR inhibitors currently undergoing clinical or preclinical evaluation (Cuthbertson et al, 2021). In ALL, methotrexate treatment has a high initial success rate, but therapy-induced selection of drug-resistant cells decreases its therapeutic index, which involves depletion of FPGS activity in at least 8% of pediatric ALL patients (Nguyen et al, 2008; Li et al, 2020). Accordingly, polyglutamylation-independent, non-classical antifolates were acknowledged as potential agents to overcome methotrexate resistance (Sohn et al, 2004; Cho et al, 2007; Kim et al, 2013). Here, we report on the characterization of C1, a novel polyglutamylation-independent antifolate. C1 has been studied as a pesticide (patent WO9820878A1) but recently regained interest for use in human subjects after showing potent inhibition of tumor growth and limited toxicity in a mouse model, where it was studied for potential effects on growth hormone signaling (Van der Velden et al, 2022). To our knowledge, our study is the first in-depth characterization of C1 as antifolate in human tissue–derived model systems, although a group of compounds with similarities to C1 has been studied in vitro using purified hDHFR (Algul et al, 2011). We find that C1, like methotrexate, selectively binds and inhibits DHFR, impairs one-carbon transfer to anabolic reactions and inhibits cellular growth. In contrast to methotrexate, C1 displays strongest DHFR inhibition in FPGS-deficient cellular contexts and outperforms the FPGS-independent antifolate trimetrexate in polyglutamylation-deficient tumor organoids.

To interrogate the cellular interaction partners of C1, we used TPP (Molina et al, 2013; Savitski et al, 2014; Mateus et al, 2020a, 2020b). The results unambiguously identify DHFR as the primary target of C1. Furthermore, TPP of C1-treated cells revealed destabilization of multiple one-carbon cycle enzymes, presumably through decreased substrate availability as a result of THF depletion. These observations are in accordance with C1, acting as an antimetabolite and indicate that the folate pathway is the main cellular process affected by C1 intervention. In addition, our findings support the use of TPP for identification of protein engagement by cellular metabolites (Huber et al, 2015). By using a modified CETSA protocol similar to two-dimensional TPP (Becher et al, 2016), we performed semi-quantitative measurements of intracellular DHFR-antifolate complexes. This approach allowed us to demonstrate that C1, despite having a lower affinity for DHFR in vitro, may inhibit DHFR more potently than methotrexate in cells due to efficient cellular entry, resulting from its lipophilic nature. In line with this assumption, C1 concentrations needed for optimal intracellular DHFR stabilization ranged from 10 to 100 nM, matching the 28 nM $K_d$ that we determined for the DHFR–C1 complex by ITC. By contrast, 100 nM methotrexate was needed for intracellular complex formation with DHFR, an order of magnitude increase over the 5 nM Kd determined for the DHFR–methotrexate complex by ITC, thus showing that its efficacy is limited by cellular import and retention. These results suggest that FPGS deficiency in tumor cells will further impair the intracellular retention of methotrexate, whereas at the same time lowering intracellular folate concentrations and thus create a vulnerability for DHFR inhibition by C1.

Loss of FPGS expression links to treatment resistance to 5-FU (Sohn et al, 2004), the most commonly applied chemotherapy in CRC. 5-FU efficacy typically depends on ternary complex formation with TYMS and 5,10-methylene–THF (Longley et al, 2003). 5-FU–mediated complex formation is compromised upon FPGS depletion due to a decrease in intracellular THF levels (Longley et al, 2003; Sohn et al, 2004), thus offering a clinically relevant condition to investigate whether polyglutamylation-deficient cancer cells display an acquired sensitivity to C1. To obtain proof-of-concept, we used patient-derived CRC organoids characterized by FPGS deficiency or FPGS overexpression. Our results confirm that FPGS-deficient CRC organoids are relatively insensitive to 5-FU, methotrexate, and trimetrexate but acquire a vulnerability for treatment with C1, in line with our model.

Methotrexate-resistant persister clones are cells that have an intrinsic or acquired insensitivity to therapy and are present at the start or formed during initial chemotherapy and cause relapse at later stages (Li et al, 2020). By co-culturing different organoids, we mimic the heterogeneous situation that occurs during initial chemotherapy of primary tumors in patients. Our results show that methotrexate treatment confers a competitive advantage to FPGS-deficient cells, whereas C1 treatment favors outgrowth of FPGS-overexpressing cells. This finding has two key implications. First, polyglutamylation deficiency creates vulnerability with sufficient therapeutic window for exploitation by C1. Second, FPGS-dependent and polyglutamylation-independent antifolates are not interchangeable and therapy based on a single agent will inevitably favor the outgrowth of persister clones. Our work therefore warrants a combination or alternating therapy of FPGS-dependent and polyglutamylation-independent antifolates to suppress growth of both the FPGS-deficient and FPGS-competent populations. Viability studies show that C1 has improved capabilities to inhibit expansion of FPGS-deficient CRC organoids compared with trimetrexate, the lipophilic, and FPGS-independent derivative of methotrexate. C1 and trimetrexate are lipophilic and lack a glutamic acid moiety for FPGS-mediated modification and thus both classify as non-classical antifolates. It is unclear why C1 outperforms trimetrexate, and additional biochemical studies are needed to compare relevant properties of C1 with other non-classical antifolates to address this issue, including affinity for hDHFR and subcellular distribution. C1 was found serendipitously, and we

anticipate that its structure may serve as a template for the development of improved non-classical antifolates that have optimal properties for selective inhibition of FPGS-deficient, methotrexate-resistant cells, and address the clinical need for intervention strategies with decreased risk of relapse. Recent studies revealed that a subpopulation of FPGS protein may associate with the cytoskeleton and motor proteins, allowing cells to channel folates to cellular subcompartments, depending on folate availability (Stark et al, 2021, 2023). This indicates that the function of FPGS extends beyond simply mediating intracellular accumulation of folates or classical antifolates, and additional research is required to test the effects of cytoskeleton-associated FPGS on sensitivity to chemotherapy. Apart from FPGS deficiency, methotrexate resistance develops through decreased import and increased export, for example, by inactivating mutations in SLC19A1 or overexpression of ABCG2 (Zhao & Goldman, 2003; Zarou et al, 2021). Considering the lipophilic nature of C1, it would be interesting to test if lipophilic antifolates like C1 may also selectively suppress growth of cells that are methotrexate-resistant as a result of altered membrane transport.

In conclusion, we identified a highly potent polyglutamylation-independent antifolate that selectively suppresses growth of methotrexate- or 5-FU-resistant, FPGS-deficient tumor cells. Our results show that FPGS deficiency causes an exploitable vulnerability to C1 and warrant a combination therapy of FPGS-dependent and -independent antifolates to prevent expansion of persister cells and overcome methotrexate resistance.

### Limitations of the study

This study presents the characterization of a novel, non-classical antifolate, which retains activity in polyglutamylation-deficient cells, a condition that arises due to loss of FPGS. Our work primarily focused on polyglutamylation as a determinant of sensitivity to classical and non-classical antifolates methotrexate and C1, respectively. However, other known parameters affecting antifolate efficacy, such as DHFR activity, lysosomal sequestration of folates, activity of drug efflux pumps, and differential impact on cytosolic and mitochondrial folate metabolism may contribute to differential activity of antifolate classes as well (Jansen et al, 1999; Zhao et al, 2001; Dekhne et al, 2020). In addition, FPGS-status of the model systems that we applied is based on analyses of FPGS mRNA and protein expression. Results were interpreted based on the assumption that FPGS mRNA and protein expression correlates with FPGS catalytic activity, which we did not assess experimentally.

## Materials and Methods

### Cell culture

HEK293T, HCT-116, LS 174T, and A549 were cultured in RPMI-1640 (Sigma-Aldrich) supplemented with 10% FBS (Bodinco), 2 mM UltraGlutamine (Lonza), 100 units/ml penicillin, and 100 $\mu$g/ml streptomycin (Sigma-Aldrich). Cells were cultured at 37°C in 5% $CO_2$ and regularly checked for mycoplasma.

### Organoid culture

P6T and P26T human colorectal cancer organoids were established in a previous study (Van De Wetering et al, 2015) and obtained a following material transfer agreement with Hubrecht Organoid Technology. Organoids were cultured in advanced DMEM/F12 medium (Thermo Fisher Scientific), supplemented with penicillin–streptomycin (Sigma-Aldrich), 10 mM HEPES (Thermo Fisher Scientific), 1X GlutaMAX (Thermo Fisher Scientific), B27 (Thermo Fisher Scientific), 10 mM nicotinamide (Sigma-Aldrich), 1.25 mM N-acetylcysteine (Sigma-Aldrich), 10% vol/vol Noggin-conditioned medium, 50 ng/ml human EGF (Peprotech), 500 nM A83-01 TGF-ß type 1 receptor inhibitor (Tocris), and 10 $\mu$M SB202190 P38 MAPK inhibitor (Sigma-Aldrich). Organoids were maintained in Cultrex BME (R&D systems) and dissociated using TrypLE (Thermo Fisher Scientific) during passaging. Medium was supplemented with 10 $\mu$M Y-27632 ROCK inhibitor (Selleck Chemicals) after splitting.

### Organoid electroporation

Electroporation of organoids was performed as previously described using a NEPA21 electroporator (Fujii et al, 2015). 7.2 $\mu$g PiggyBac-CMV overexpression construct, 7.2 $\mu$g rTTA-IRES-Hygro, and 5.2 $\mu$g PiggyBac transposase were used to generate organoid lines with inducible overexpression of FPGS, GGH, and empty vector. 100 $\mu$g/ml hygromycin B was used for selection of electroporated organoids.

### Organoid viability assays

For viability assays upon treatment with antifolates, advanced DMEM/F12 medium was replaced with MEM (Thermo Fisher Scientific) supplemented with 1X MEM non-essential amino acids (Thermo Fisher Scientific) and 5 $\mu$g/l vitamin B12 (Sigma-Aldrich). 1 $\mu$g/ml doxycycline was added to induce overexpression of FPGS, GGH, and empty vector constructs. Before seeding, organoids were dissociated using TrypLE (Thermo Fisher Scientific) and passed through a 100 $\mu$m cell strainer. 1000X drug dilutions were added 2 d after seeding. For monoculture viability assays, 12,500 cells/well were seeded in a 96-well plate containing 10 $\mu$l BME. CellTiter-Glo luminescent cell viability assay (Promega) was mixed at a ratio of 1:1 with culture medium, and 200 $\mu$l/well was used to quantify viability of monocultures. For co-culture viability assays, 12,500 cells/well of each line were seeded in a 96-well plate containing 50 $\mu$l BME matrix topped off with 20 $\mu$l empty BME. Outgrowth of organoids in co-culture was analyzed using a Leica THUNDER widefield microscope equipped with a 10X HC PL FLUOTAR objective (NA = 0.32) and large volume computational clearing deconvolution algorithm. Organoid surface area was quantified from Z-stack maximum projections using FIJI MorphoLibJ Morphological Segmentation plugin combined with particle analysis. Brightfield images were obtained on an EVOS M5000 imaging system (Thermo Fisher Scientific).

### Plasmids and antibodies

Human HA–DHFR was subcloned from human cDNA into pcDNA4/TO by PCR using Q5 High Fidelity 2X Mastermix (NEB). Human FPGS–FLAG

and GGH–FLAG were subcloned from cDNA clones MHS6278-202755815 (Horizon Discovery) and MHS6278-202757326 (Horizon Discovery), respectively, into pcDNA4/TO and PiggyBac–CMV–MCS–IRES–mCherry and PiggyBac–CMV–MCS–IRES–mNeonGreen by PCR using Q5 High Fidelity 2X Mastermix (NEB). PiggyBac–CMV–MCS–IRES–NLS–TagBFP was used as empty vector control. All constructs were sequence verified. The following primary antibodies were used for Western blotting (WB) and immunofluorescence (IF): rat anti-HA (11867423001; Roche), mouse anti-FLAG (F3165; Sigma-Aldrich), rabbit anti-4E-BP1 (9452; Cell Signaling), mouse anti-TOM20 (612278; BD transduction laboratories), mouse anti-LAMP1 (555798; BD Pharmingen), rabbit anti-Akt (9272; Cell Signaling), rabbit anti-FPGS (orb422877; Biorbyt), rabbit anti-vinculin (ab129002; Abcam), and mouse anti-Actin (691001; MP Biomedicals). Primary antibodies were diluted according to manufacturer's instructions. Secondary antibodies for WB and IF were diluted 1:5,000 and 1:300, respectively, and obtained from Rockland, LI-COR, or Invitrogen.

### Immunofluorescence and confocal microscopy of organoids

Organoids were released from extracellular matrix using dispase (Sigma-Aldrich) at 37°C for 30 min. Organoids were fixed in 0.1 M phosphate buffer containing 4% paraformaldehyde for 1 h. Washing and antibody stainings were performed in four-well $\mu$-slides (Ibidi), using PBS containing 0.2% Triton X-100, 1% DMSO, and 1% bovine serum albumin. 1 $\mu$g/ml DAPI was used to stain nuclei. Organoids were mounted using mounting medium (Ibidi) and analyzed using a Zeiss LSM700 confocal microscope equipped with 63X plan-apochromat oil immersion objective (NA = 1.40).

### Cell viability assays

5,000 cells/well were seeded in a 96-well plate and drugs diluted in full culture medium were added the next day. Viability was quantified with CellTiter-Glo luminescent cell viability assay (Promega) according to manufacturer's instructions. Luciferase activity was measured on a Berthold Centro LB960 luminometer. Drug sensitivity of a panel of cancer cell lines for C1 was determined at OncoLead, using a sulforhodamine B viability assay after 72 h of treatment (Figs 1B and S5 and Table S1).

### Drugs and rescue metabolites

Pyrimethamine (Sigma-Aldrich), methotrexate (Selleck Chemicals), trimethoprim (Sigma-Aldrich), trimetrexate hydrochloride (CI-898; Santa Cruz) were dissolved in DMSO. Rescue agents were dissolved in water unless indicated otherwise and treatments with hypoxanthine (dissolved in 67% formate; Sigma-Aldrich), folinic acid (5-formyl-THF; Sigma-Aldrich), thymidine (Sigma-Aldrich), glycine (Sigma-Aldrich), adenosine (Sigma-Aldrich), and guanosine (Sigma-Aldrich) were performed as indicated. Compound C1 was synthesized by Specs Compound Handling as described in patent WO2021078995A1.

### RT-qPCR

RT-qPCR was performed as previously described (Omerzu et al, 2019). RNA was isolated using a RNeasy Mini Kit (QIAGEN). Turbo DNAse (Thermo Fisher Scientific) was used to remove genomic DNA. cDNA was synthesized using a iScript cDNA synthesis kit (Bio-Rad). Human *FPGS* and *GGH* primers were previously published (Driehuis et al, 2020). Other primers used: CTTTTGCGTCGCCAG (*GAPDH* forward), TTGATGGCAACAATATCCAC (*GAPDH* reverse). *FPGS* and *GGH* expression was calculated relative to *GAPDH* using the $2^{-\Delta Ct}$ method.

### CETSAs

HEK293T cells were cultured in RPMI-1640 to 50% confluency in 100 mm plates and subsequently transfected with 6 $\mu$g/dish pcDNA4/TO_HA-DHFR or 5 $\mu$g/dish pcDNA4/TO_HA-DHFR + 5 $\mu$g/dish pcDNA4/TO_FPGS-FLAG. Transfections were performed using polyethylenimine (PEI), and medium was refreshed after 4 h incubation with the transfection mix. CETSA was performed as described before (Zheng et al, 2018) and samples were heated to 37, 39, 42.3, 46.4, 51.9, 56.1, 59, and 61°C in an S1000 thermal cycler (Bio-Rad). For CETSA with dose-response, drug incubations were performed in PCR tubes, using 100 $\mu$l cell suspension per condition. 100X drug stocks were diluted to 1X and incubated for 4 h. CETSA results were analyzed by mixing the cell extracts with sample buffer and subsequent WB. HA-DHFR stabilization was quantified by calculating signal intensity of HA over Actin (HA/Actin). Z-score transformation was used to normalize HA/actin signal intensity per drug (C1, methotrexate, or pyrimethamine) and cell line (overexpressing mTurquoise2 or FPGS–FLAG).

### Cell lysis and WB

For organoids, extracellular matrix was removed using Cell Recovery Solution (Corning). Cells and organoids were washed once with ice-cold PBS and lysed in RIPA buffer (25 mM Tris, pH 7.6, 150 mM NaCl, 1% Nonidet P40 substitute [Sigma-Aldrich], 1% sodium deoxycholate, 0.1% SDS, 50 mM NaF, 1 mM PMSF, 10 $\mu$g/ml leupeptin, 10 $\mu$g/ml aprotinin). Lysates were centrifuged at 15,000$g$ for 15 min at 4°C. Soluble fraction was isolated and mixed with 5X sample buffer (350 mM Tris, pH 6.8, 10% SDS, 20% glycerol, 2.5% 2-mercaptoethanol, 0.025% bromophenol blue) to a final concentration of 1X. Samples were denatured at 95°C for 10 min. WB was performed under standard procedures using SDS–PAGE to resolve samples, followed by transfer to Immobilon-FL PVDF membrane (Millipore). Membranes were blocked with Odyssey blocking buffer (LI-COR) diluted 1:1 in TBS. Primary and secondary antibodies were diluted in TBS + 0.05% Tween 20. Membranes were imaged on an Amersham Typhoon NIR laser scanner (GE Healthcare).

### TPP

TPP was performed as previously described (Becher et al, 2018). In brief, cells were harvested after 1 h of treatment with 10 $\mu$M compound C1 or DMSO, washed with PBS and 10 aliquots, each of

$1 \times 10^6$ cells in 100 $\mu$l PBS, were distributed in a 96-well PCR plate. After centrifugation (300$g$ for 3 min) and removal of most of the supernatant (80 $\mu$l), each aliquot was heated for 3 min to a different temperature (37°C, 40.4°C, 44°C, 46.9°C, 49.8°C, 52.9°C, 55.5°C, 68.6°C, 62°C, and 66.3°C) in a PCR machine (8800; Agilent SureCycler) followed by 3 min at RT. Cells were lysed with 30 $\mu$l ice-cold lysis buffer (final concentration 0.8% NP-40, 1.5 mM MgCl2, protease inhibitors, phosphatase inhibitors, 0.4 U/$\mu$l benzonase) on an orbital plate shaker (500 rpm) at 4°C for 1 h. The PCR plate was then centrifuged at 300$g$ for 3 min at 4°C to remove cell debris, and the supernatant was filtered at 300$g$ for 3 min at 4°C through a 0.45-$\mu$m 96-well filter plate (MSHVN4550; Millipore) to remove protein aggregates. Of the flow-through, 25 $\mu$l was mixed with 2× sample buffer (180 mM Tris, pH 6.8, 4% SDS, 20% glycerol, 0.1 g bromophenol blue) and kept at −20°C until prepared for mass spectrometry analysis, whereas the remainder was used in a BCA (Thermo Fisher Scientific), to determine the protein concentration. Samples were diluted to 1 $\mu$g/$\mu$l in 1x sample buffer based on the protein concentrations in the lowest two temperatures (37°C and 40.4°C).

### MS sample preparation and measurement

Proteins were digested as previously described (Mateus et al, 2020a, 2020b). Briefly, 10 $\mu$g of protein (based on the protein concentrations in the lowest two temperatures) was added to a bead suspension (10 $\mu$g of beads [Sera-Mag Speed Beads, 4515-2105-050250, 6515-2105-050250; Thermo Fischer Scientific] in 10 $\mu$l 15% formic acid and 30 $\mu$l ethanol) and incubated on an orbital plate shaker (500 rpm) for 15 min at RT. Beads were washed four times with 70% ethanol, and proteins were digested overnight in 40 $\mu$l digest solution (5 mM chloroacetamide, 1.25 mM TCEP, 200 ng trypsin, and 200 ng LysC in 100 mM HEPES pH 8). Peptides were then eluted from the beads, vacuum-dried, reconstituted in 10 $\mu$l of water, and labeled for 1 h at RT with 18 $\mu$g of TMT10plex (Thermo Fisher Scientific) dissolved in 4 $\mu$l of acetonitrile (the label used for each experiment can be found in Table S3). The reaction was quenched with 4 $\mu$l of 5% hydroxylamine, and samples were combined by temperature. Samples were acidified and desalted using StageTips (Rappsilber et al, 2007) and eluted with 2 × 30 $\mu$l of buffer B (80% acetonitrile, 0.01% TFA). Samples were fractionated using the Pierce High pH Reversed-Phase Peptide Fractionation Kit (Thermo Fisher Scientific) into three fractions (Fraction No. 4, 7, and 8). The flow-through, wash, and TMT wash fractions were pooled together with fraction 4. Peptides were applied to reverse-phase chromatography using a nanoLC-Easy1000 coupled online to a Thermo Orbitrap Q-Exactive HF-X. Using a 120 min gradient of buffer B, peptides were eluted and subjected to tandem mass spectrometry. The mass spectrometer was operated in Top 20 mode and dynamic exclusion was applied for 30 s.

### MS data analysis

MS data were analyzed using Proteome Discoverer (version 2.2; Thermo Fisher Scientific). Data were searched against the human UniProt database. Search parameters: trypsin, missed cleavages 3, peptide tolerance 10 ppm, 0.02 D for MS/MS tolerance. Fixed modifications were carbamidomethyl on cysteines and TMT10plex on lysine. Variable modifications included acetylation on protein N terminus, oxidation of methionine, and TMT10plex on peptide N-termini.

### Abundance and stability score calculation

The Proteome Discoverer output files were loaded into R, merged, filtered for duplicates, and proteins with less than two unique peptides and saved in an ExpressionSet R-object. Potential batch effects were removed using limma (Ritchie et al, 2015), and data were normalized using variance stabilization, vsn strategy (Huber et al, 2002). Normalization was performed for each temperature independently, to account for the decreasing signal intensity at the higher temperatures. The abundance score of each protein was calculated as the average log2 fold change at the two lowest temperatures (37°C and 40.4°C). The stability score of each protein was calculated by subtracting the abundance score from the log2 fold changes of all temperatures and calculating the sum of the resulting values. To assess the significance of abundance and thermal stability scores, we used a limma analysis, followed by an FDR analysis using the fdrtool package.

### PANTHER statistical overrepresentation analysis

From the thermal proteome profile of C1-treated LS 174T cells, all proteins displaying significant thermal shifts were used as input for PANTHER (version 16.0, released 2020-12-01) statistical overrepresentation test (released 20210224) as previously described (Mi et al, 2019). Binomial test with FDR correction was used. PANTHER GO-Slim Biological Process and PANTHER pathways were used as annotation datasets. Hierarchical clustering was applied to the PANTHER GO-Slim annotated dataset to filter out the most specific subclasses.

### Kinome-wide screening for kinase inhibition

In vitro screening for kinase inhibition by compound C1 was performed by Thermo Fisher Scientific. For the LanthaScreen kinase activity assay, fluorescein-labeled substrate was incubated with a kinase of interest and ATP to allow kinase-dependent phosphorylation. A terbium-labeled antibody was subsequently used to detect substrate phosphorylation, resulting in Förster resonance energy transfer (FRET). Time-resolved (TR) FRET was interpreted as a readout of kinase activity. For the Adapta kinase activity assay, kinase activity was reconstituted in vitro, followed by ADP detection using an europium-labeled antibody and an AlexaFluor647-labeled ADP-tracer, resulting in FRET in the absence of ADP production. Kinase activity produces unlabeled ADP, resulting in displacement of the labeled ADP-tracer and TR-FRET inhibition. Increased TR-FRET was interpreted as a readout for kinase inhibition. For the Z'LYTE assay, kinase activity was reconstituted in vitro using fluorescein- and coumarin-labeled FRET peptides as substrates. A protease cleaving non-phosphorylated peptides was subsequently added to the reaction. Proteolytic cleavage of the substrate peptides disrupts FRET and FRET inhibition was interpreted as a readout for kinase inhibition. Results were combined and visualized using Coral (Metz et al, 2018).

### In vitro DHFR activity assay

In vitro hDHFR activity was analyzed by using a colorimetric assay kit according to manufacturer's instructions (Sigma-Aldrich) to quantify hDHFR-dependent NADPH turnover by monitoring absorbance at 340 nm. Briefly, C1 and methotrexate were dissolved to 10 mM in 30% acetonitrile + 0.1% formic acid. Drug stocks were diluted in assay buffer to final concentrations of 10–10,000 nM. Absorbance at 340 nm was measured using a SmartSpec spectrophotometer (Bio-Rad) every 60 s for 5 min and quantified using kinetics software (Bio-Rad).

### Statistical analysis, correlation analysis, and curve fitting

Statistical tests were performed as indicated in figure legends. All statistical analyses were performed in GraphPad Prism 9. D'Agostino and Pearson's test was used to test for normality and lognormality. In case of small sample sizes, descriptive statistics were used to compare SDs for statistical tests that assume equal SDs. Binding and viability curves were fitted in GraphPad Prism 9, using a nonlinear regression asymmetrical sigmoidal (five parameter) model where X is concentration. Correlation between gene expression and drug sensitivity was calculated using Spearman correlation coefficient. Gene expression levels and drug sensitivity data for methotrexate and pyrimethamine were retrieved from the Genomics of Drug Sensitivity in Cancer database and the Cancer Cell Line Encyclopedia (Ghandi et al, 2019).

### ITC

ITC measurements were performed in a Low Volume NanoITC (TA Instruments-Waters LLC). hDHFR (4.3 $\mu$M) (8456-DR-100; R&D Systems) was prepared in assay buffer (50 mM MES hydrate, 25 mM Tris, 100 mM Nacl, 25 mM ethanolamine, 2 mM DTT). C1 (1 $\mu$M), methotrexate (1 $\mu$M), and $\beta$-NADPH (125 $\mu$M) were also prepared in the same buffer. 300 $\mu$l of the C1 or methotrexate in presence or absence of $\beta$-NADPH was present in the cell (maximum cell volume is 169 $\mu$l), and 50 $\mu$l of hDHFR was loaded in the syringe as titrant. At 37°C, 2 $\mu$l of hDHFR was injected into the cell every 300 s, except the first injection which is of 0.96 $\mu$l. All experiments were performed at 37°C while stirring at 300 rpm. The data were analyzed with the NanoAnalyze Software (TA instruments), and background titration of hDHFR to buffer is subtracted from all thermograms.

### Determination of distribution coefficient (logD)

The distribution coefficient (log$D$) was determined for C1 at various aqueous phase pH values by using the shake-flask procedure as previously described (Andrés et al, 2015).

### Generation of docking models

The protocol we followed for this system is based on our, recently published, protein-small molecule shape-restrained docking protocol (Koukos et al, 2021). For in-depth details regarding the pre-processing and docking protocol see the "Materials and Methods"

section of the original publication. The main steps are summarized in the sections below.

### Template identification

First, we searched the Protein Data Bank (PDB) (Berman et al, 2000), using the SMILES string of the target compound and the FASTA sequence of the target receptor as inputs, and in addition filtering for receptors with at least one co-crystallised compound (Weininger, 1988; Weininger et al, 1989). After removing unsuitable templates (low-resolution structures, receptors only crystallised with crystallisation buffers, etc.), we extracted the template SMILES strings. Using the extracted template SMILES strings and the target compound C1 SMILES string as inputs, we compared the chemical similarity of the target compound C1 to all template compounds. For this similarity comparison, we computed the Tversky similarity (biased with a weight of 0.8 for the target compound vs one of 0.2 for the template compounds) over the maximum common substructure (MCS) as identified with the rdFMCS implementation of RDKit (version 2020.09.3) (Tversky, 1977). After calculating all similarity values, we ranked the compounds by their Tversky similarity and selected the one with the highest value (closer to 1 as opposed to 0). For this modelling effort, we opted to perform the docking using the template identified via the above procedure (PDB ID: 5HPB) and the second-best template (PDB ID: 4KBN).

### Conformer generation

Because no 3D structures of the C1 compound were available, we opted to generate 3D conformers of the compound starting from its isomeric SMILES string: CC1 = C(C(=NC(=N1)N)N)/C=C/C2(CC2) C3 = CC = C(C=C3)C(F)(F)F. In total, 64 conformers were generated with RDKit using the 2020 parameter set although making use of energy minimisation and the ETKDG algorithm (Riniker & Landrum, 2015; Wang et al, 2020). We provide the ensemble of generated conformers to HADDOCK without any additional filtering.

### System preparation

We removed all crystallographic waters and all crystallisation artefacts from both template receptor structures before docking, whereas maintaining the relevant NADPH cofactor. Topologies and parameters for the cofactor and compound C1 were generated with PRODRG (version 070118.0614) (Schüttelkopf & Van Aalten, 2004).

### Generation of shape-based restraints

All docking simulations were carried out with the command-line version of HADDOCK 2.4 (January 2021 release), our integrative modelling platform, using in-house computational resources (Dominguez et al, 2003; van Zundert et al, 2016). HADDOCK integrates experimentally (or otherwise) obtained data in the docking to guide the simulation toward generating poses that satisfy the provided data. For this modelling effort, we made use of restraints extracted from the shape of the template compounds. Specifically, after identifying the template receptors via the procedure laid out above,

we transformed the heavy atoms of the template compounds into dummy beads and then defined ambiguous distance restraints with an upper limit of 1 Å between the shape beads and the non-hydrogen atoms of the compound to be docked. These restraints are always defined from the smaller to the larger body, so for the docking based on the 5HPB template, the restraints were defined from the shape beads to any non-hydrogen atom of the compound, whereas for the one based on 4KBN they were defined from the non-hydrogen compound atoms to any shape bead. In addition to these shape-based restraints, we also defined restraints between the non-hydrogen atoms of the NADPH cofactor and its surrounding residues meant to maintain the original geometry of the cofactor relative to its surroundings.

### Docking

For the docking, we generated 1,280 models during the rigid-body stage ($20 * N_{generated conformers}$) out of which the top 200 proceeded to the flexible refinement stage. We set the total number of components active during docking to 3 (the template receptor, the generated conformers, and the template-based shape), disabled the systematic sampling of 180-symmetrical poses during the rigid-body stage, disabled the random removal of restraints, fixed the position of the template receptor, and shape to their original positions and disabled the deletion of non-polar hydrogens. We used constant dielectric for both stages and set its value for the refinement stage to 10. We lowered the scaling of intermolecular interactions during rigid-body minimisation to 0.001 of its original value to allow the generated conformers to more easily penetrate into the binding pocket and also ignored the contribution of the vdW term during the scoring of the poses for the rigid body stage. We clustered the generated models using an RMSD cut-off value of 1.5 Å.

### Scoring

The scoring functions used for both stages are:

$HS - rigid\ body = 0.0*E_{vdw} + 1.0*E_{elec} + 1.0*E_{desolv} + 0.01*E_{AIR} - 0.01*BSA,$

$HS - refinement = 1.0*E_{vdw} + 1.0*E_{elec} + 1.0*E_{desolv} + 0.1*E_{AIR} - 0.01*BSA,$

where HS stands for HADDOCK score, Evdw, Eelec, and Edesolv stand for van der Waals, Coulomb electrostatics, and desolvation energies, respectively, and BSA for the buried surface area. The non-bonded components of the score (Evdw, Eelec) are calculated with the OPLS forcefield (Jorgensen & Tirado-Rives, 1988). The desolvation energy is a solvent accessible surface area–dependent empirical term which estimates the energetic gain or penalty of burying specific sidechains upon complex formation (Fernández-Recio et al, 2004).

## Data Availability

The mass spectrometry proteomics datasets have been deposited in the ProteomeXchange Consortium via the PRIDE partner repository (https://www.ebi.ac.uk/pride/) (Perez-Riverol et al, 2019) under the dataset identifier PXD040155.

## Supplementary Information

## Acknowledgements

We thank our colleagues of the MM Maurice laboratory and Center for Molecular Medicine for fruitful discussions, feedback, and suggestions; Corlinda ten Brink for assistance during imaging experiments using Cell Microscopy Core equipment; Joep Sprangers and Remco Sleiderink for providing the PiggyBac–CMV–MCS–IRES–mNeonGreen plasmid; Dennis Piet, Sirik Deerenberg, and Tom Speksnijder for the synthesis of C1; Nanda Sprenkels for QC, formulation, and solubility studies on C1; Ingrid Jordens for providing normal colon organoids (P26N) RNA; Michael Hadders, Ingrid Jordens and Joep Sprangers for critical reading of the article; the members of the NWO-TTW user committee for their critical input and discussions. This work is part of the Oncode Institute, which is partly financed by the Dutch Cancer Society (KWF). This work was supported by the Netherlands Organization for Scientific Research (NWO) domain TTW, grant 16083 (J Klumperman, GJ Strous, and JA Mol), Zon-MW VICI grant 91815604 (to MM Maurice), Zon-MW TOP grant 91218050 (MM Maurice), and Gravitation project IMAGINE! (MM Maurice).

### Author Contributions

F van der Krift: conceptualization, formal analysis, investigation, visualization, methodology, and writing—original draft, review, and editing.
DW Zijlmans, R Shukla, PI Koukos: conceptualization, formal analysis, investigation, visualization, methodology, and writing—original draft.
A Javed: formal analysis, investigation, visualization, and methodology.
LLE Schwarz, D Gahtory, and M van den Nieuwboer: formal analysis, investigation, and methodology.
EPM Timmermans-Sprang: conceptualization, formal analysis, investigation, visualization, and methodology.
PEM Maas: resources.
JA Mol: conceptualization, funding acquisition, and investigation.
GJ Strous: conceptualization and funding acquisition.
AMJJ Bonvin: conceptualization, supervision, investigation, and methodology.
M van der Stelt: conceptualization, investigation, and methodology.
EJA Veldhuizen, M Weingarth, and M Vermeulen: formal analysis, supervision, investigation, and methodology.
J Klumperman: conceptualization, formal analysis, supervision, funding acquisition, investigation, and methodology.
MM Maurice: conceptualization, formal analysis, supervision, funding acquisition, investigation, visualization, methodology, and writing—original draft, review, and editing.

### Conflict of Interest Statement

MM Maurice is an inventor on patents related to membrane protein degradation; she is co-founder and shareholder of Laigo Bio.

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
