## [Reviewer comments · Life Science Alliance]

Life Science Alliance

A novel antifolate suppresses growth of FPGS-deficient cells and overcomes methotrexate resistance

Felix van der Krift, Dick Zijlmans, Rhythm Shukla, Ali Javed, Panagiotis Koukos, Laura Schwarz, Elpetra Timmermans-Sprang, Peter Maas, Digvijay Gahtory, Maurits van den Nieuwboer, Jan Mol, Ger Strous, Alexandre Bonvin, Mario van der Stelt, Edwin Veldhuizen, Markus Weingarh, Michiel Vermeulen, Judith Klumperman, and Madelon Maurice

DOI: <https://doi.org/10.26508/lsa.202302058>

Corresponding author(s): Madelon Maurice, University Medical Center Utrecht

Review Timeline:

Submission Date:	2023-03-24
Editorial Decision:	2023-04-21
Revision Received:	2023-07-17
Editorial Decision:	2023-07-31
Revision Received:	2023-08-07
Accepted:	2023-08-07

Transaction Report:

April 21, 2023

Re: Life Science Alliance manuscript #LSA-2023-02058-T

Madelon M. Maurice
University Medical Center Utrecht
Cell Biology
Heidelberglaan 100
Utrecht 3584CX
Netherlands

Dear Dr. Maurice,

Thank you for submitting your manuscript entitled "A novel antifolate suppresses growth of FPGS-deficient cells and overcomes methotrexate resistance" to Life Science Alliance. The manuscript was assessed by expert reviewers, whose comments are appended to this letter. We invite you to submit a revised manuscript addressing the Reviewer comments.

Thank you for this interesting contribution to Life Science Alliance. We are looking forward to receiving your revised manuscript.

Sincerely,

B. MANUSCRIPT ORGANIZATION AND FORMATTING:

Reviewer #1 (Comments to the Authors (Required)):

In this manuscript, the authors described the studies of a novel non-classical antifolate, C1. Based on a previous screen, the authors identified C1 as a DHFR inhibitor. The authors studied and compared the DHFR activities of C1 and methotrexate, a classical antifolate. To further elucidate the mechanism behind the differential activities, the authors conducted modeling studies and tested both C1 and methotrexate in FPGS deficient cell lines. The authors demonstrated that the C1 possessed superior efficacy in FPGS deficient cell lines and tissue organoids. Overall, the experiments were well designed and the conclusions were strongly supported by the results.

Overall, I consider the manuscript a significant advance in the field of antifolate development. I would suggest the following revisions to improve the manuscript:

1. In general, the manuscript lacks discussions and descriptions on non-classical antifolates. Since you are studying a non-classical antifolate, please discuss how your discovery can address any unmet needs or help develop better non-classical antifolates.
2. For the modeling study, please provide quantitative measures of the bound complexes (such as calculated binding free energy) in order to better explain the observed difference in in vitro binding affinity.
3. In Figure 4C Panel 1, there seems to be a large discrepancy between the duplicates in DHFR stabilization assay in HEK293T treated by C1, please briefly discuss.

Reviewer #2 (Comments to the Authors (Required)):

This study reports the extensive characterization of a novel lipophilic antifolate C1 which has dihydrofolate reductase (DHFR) as its primary target. Studies including in vitro cancer cell line models and colon cancer organoids demonstrated enhanced potency of C1 to cells with relative low folylpolyglutamate synthetase (FPGS) expression.

General comment

For decades differential activities of FPGS in tumor vs normal tissues has set the rationale for cancer chemotherapy with high dose MTX therapy and leucovorin rescue, and also for provoking enhanced sensitivity to lipophilic antifolate drugs. Overall, data from this study are consistent with also C1 eliciting enhanced sensitivity to cancer/organoid cells with relatively lower FPGS expression. As such, these data for C1 as a lead compound are of interest for follow up preclinical studies. In this regard, the results of this study may not have revealed the full potential of C1 because experimental conditions did not always take into account known properties of lipophilic vs polyglutamatable antifolates. A few of these will be indicated below and are advised to be discussed in the Discussion section as potential limitations of this study or, if possible, experimentally addressed.

Specific comments:

Abstract, line 31 and Introduction, 67: FPGS is an ATP-dependent enzyme, thus a synthetase. Folylpolyglutamate synthase should be corrected to folylpolyglutamate synthetase.

This study mainly focused on FPGS and GGH (mRNA) expression in relation to potency of C1. However, early studies for DHFR targeted lipophilic antifolates trimetrexate and pyrimethamine pinpointed at intracellular folate status, DHFR activity, lysosomal sequestration, being substrate for the drug efflux transporter ABCB1/P-Glycoprotein, and ability to impact cytoplasmic and/or mitochondrial folate metabolism as additional parameters determining their potency (PMID: 11274972, 10101035, 31707355). Other than folate status, these parameters were not considered for C1 in experimental settings with cell lines and organoids. What is the logP for C1? These points should be discussed as a limitation.

Given the multiple parameters contributing C1 potency, it may not be surprising that drug screening of C1 in a panel of diverse cell lines with variability in cell doubling times revealed a wide range of C1 drug sensitivities in a 72 hr experiment (Fig 1B, Supple Table 1). Moreover, SRB analysis as a readout system is suboptimal for testing antifolate compounds since antifolate-induced folate deficiency comes with macrocytic cells with more protein, leading to under-interpretation of drug effects and IC50 values. This is typically manifested by dose response curves that flatten well above zero relative viability (noted in Fig 2D/F, Suppl Fig 2).

What was the rationale to select A549 cells for experiments described in (Suppl) Figure 2?

Lines 239-280, experiments Figure 4. There is no validation that FPGS transfection in HEK293T is accompanied with increased FPGS catalytic activity and/or increased MTX-polyglutamate accumulation. Also, the remark (line 268-269) that 10 nM of MTX does not lead to intracellular DHFR stabilization is fully explainable by experimental conditions settings. Uptake of MTX via folate transporters is suboptimal at 100 nM and with a K_m of FPGS for MTX of 100 μ M, a 90 min exposure time is too short to induce similar levels of DHFR stabilization as for C1 being transport and polyglutamylation independent.

Organoid experiments and FPGS/GGH transfections (Fig 5, Suppl Fig 5). Based on results for (only) 1 - 2 organoids cultures, the outcome should not be overinterpreted. The authors show only relative mRNA expression data for FPGS and GGH, and no actual catalytic activities. Studies by Stark et al (in ref list) reported no apparent correlation for FPGS mRNA and FPGS enzyme activities. Moreover, cellular FPGS enzyme activities may vary by 2-3 orders of magnitude, being highest in highly proliferative cancer cells (PMID: 1435744). Finally, some recent novel functions of FPGS were described that may also be of relevance for this study (PMID: 33676037, 36721160).

Reviewer 1

“In general, the manuscript lacks discussions and descriptions on non-classical antifolates. Since you are studying a non-classical antifolate, please discuss how your discovery can address any unmet needs or help develop better non-classical antifolates.”

We thank the reviewer for bringing up this important point. We have now addressed this point by describing C1 in more detail as a non-classical antifolate (e.g. lines 239-244 & 417-419). In addition, we revised the discussion section to better clarify how our discovery may address a clinical need for non-classical antifolates (lines 421-424).

“For the modeling study, please provide quantitative measures of the bound complexes (such as calculated binding free energy) in order to better explain the observed difference in in vitro binding affinity.”

We thank the reviewer for this interesting thought. To our knowledge, there are currently no reliable computational methods available to accurately predict binding affinity, except for methods that require tremendous computational power (reviewed in e.g. PMID: 30061498). The docking models presented in the manuscript represent poses with highest scores, meaning that these poses are predicted to be most energetically favorable. However, due to the simplifications required for *in silico* evaluation of a large number of docking poses, it is not possible to accurately predict and compare binding affinities from these models. We therefore interpreted the models put forth by the docking approach in light of our experimental data obtained *in vitro* (isothermal titration calorimetry and enzymatic assays), which more accurately describe the differences in affinity.

“In Figure 4C Panel 1, there seems to be a large discrepancy between the duplicates in DHFR stabilization assay in HEK293T treated by C1, please briefly discuss.”

We agree with the reviewer that these results were not optimally displayed. To address this point, we repeated the experiments and optimized the quantification method used in Figure 4C of the original manuscript. We realized that normalization of DHFR-HA intensity towards the DHFR-HA signal in DMSO-treated cells is not optimal due to low-intensity signals in the absence of drug-induced DHFR stabilization. We therefore revised the quantification method by i) introducing a loading control (Actin) and calculating a ratio of HA over Actin (HA/Actin), and ii) by expressing HA/Actin as a Z-score-transformed value, calculated per cell line and for each drug treatment condition. With these adaptations we were able to reveal more accurately at which drug concentration maximal DHFR-HA stabilization was achieved. Results are displayed in **new Figure 4** and **new Supplementary Figure 6**.

We thus addressed this point by modification of the experimental protocol and repeating of the entire set of experiments – which also allowed us to address a comment by reviewer 2, discussed below.

Reviewer 2

“Abstract, line 31 and Introduction, 67: FPGS is an ATP-dependent enzyme, thus a synthetase. Folylpolyglutamate synthase should be corrected to folylpolyglutamate synthetase.”

We thank the reviewer for pointing out that we wrongfully called FPGS a synthase and corrected the errors accordingly.

“This study mainly focused on FPGS and GGH (mRNA) expression in relation to potency of C1. However, early studies for DHFR targeted lipophilic antifolates trimetrexate and pyrimethamine pinpointed at intracellular folate status, DHFR activity, lysosomal sequestration, being substrate for the drug efflux transporter ABCB1/P-Glycoprotein, and ability to impact cytoplasmic and/or mitochondrial folate metabolism as additional parameters determining their potency (PMID: 11274972, 10101035, 31707355). Other than folate status, these parameters were not considered for C1 in experimental settings with cell lines and organoids. What is the logP for C1? These points should be discussed as a limitation.”

We fully agree with the reviewer that, besides polyglutamylolation, multiple additional parameters affect cellular sensitivity to classical and non-classical antifolates. We added a paragraph to the discussion section to discuss limitations of the study and state that our study is limited to the effects of FPGS-mediated polyglutamylolation on sensitivity to C1 and methotrexate. To address the question about C1's logP, we experimentally determined the logD for C1. Results are included in **new Supplementary Figure 4**. The results show that C1 is a lipophilic compound at physiological pH and that its lipophilicity decreases in acidic conditions.

“Given the multiple parameters contributing C1 potency, it may not be surprising that drug screening of C1 in a panel of diverse cell lines with variability in cell doubling times revealed a wide range of C1 drug sensitivities in a 72 hr experiment (Fig 1B, Supple Table 1). Moreover, SRB analysis as a readout system is suboptimal for testing antifolate compounds since antifolate-induced folate deficiency comes with macrocytic cells with more protein, leading to under-interpretation of drug effects and IC50 values. This is typically manifested by dose response curves that flatten well above zero relative viability (noted in Fig 2D/F, Suppl Fig 2). What was the rationale to select A549 cells for experiments described in (Suppl) Figure 2?”

Please note that Sulforhodamine B analysis was only used for the viability assays in Figure 1B and Supplementary Table 1. Other viability experiments shown in the manuscript (including those in Figure 2D, 2F, 5B and S3) were analyzed either with imaging-based quantifications of organoid outgrowth (Figure 5A, 5C, 5D), or using the CellTiter-Glo viability assay (Promega). CellTiter-Glo is a luciferase-based viability assay used to quantify metabolically active cells, and depends on cell-derived ATP. Using CellTiter-Glo assays, we observed that antifolates cause cytostatic rather than cytotoxic effects. This means that, even at higher drug concentrations, the cultures still contain viable cells. Furthermore, arrested cells still produce ATP, which also explains the observation that the viability curves flatten well above zero. We selected A549 cells for the experiments in Figure 2 because these cells are highly sensitive to C1 (see Supplementary Table 1) and because they were previously used to assess the effects of antifolate treatment on mTORC1 signaling (PMID: 29091770), similar to our experiments in Supplementary Figure 3B.

“Lines 239-280, experiments Figure 4. There is no validation that FPGS transfection in HEK293T is accompanied with increased FPGS catalytic activity and/or increased MTX-polyglutamate accumulation. Also, the remark (line 268-269) that 10 nM of MTX does not lead to intracellular DHFR stabilization is fully explainable by experimental conditions settings. Uptake of MTX via folate transporters is suboptimal at 100 nM and with a Km of FPGS for MTX of 100 μM, a 90 min exposure time is too short to induce similar levels of DHFR stabilization as for C1 being transport and polyglutamylolation independent.”

We thank the reviewer for bringing up this important point. We did confirm overexpression of FPGS protein by Western-blotting (**new Supplementary Figure 6B**), but indeed did not experimentally assess whether this leads to an increase in FPGS catalytic activity. We now added a comment and acknowledge this point as a limitation of our study in the discussion section.

We thank the reviewer for pointing out the important point that a 90-min exposure time is too short because of suboptimal uptake and polyglutamylation of methotrexate at 100 μ M. To address this issue, we repeated the entire experiment twice with a prolonged drug treatment time of 4 hours. Additionally, we improved the quantification method (based on a suggestion of reviewer 1). We added this new set of results to the manuscript (**new Figure 4, new Supplementary figure 6C**) and adapted our conclusions (lines 260-279 and 385-395).

“Organoid experiments and FPGS/GGH transfections (Fig 5, Suppl Fig 5). Based on results for (only) 1 - 2 organoids cultures, the outcome should not be overinterpreted. The authors show only relative mRNA expression data for FPGS and GGH, and no actual catalytic activities. Studies by Stark et al (in ref list) reported no apparent correlation for FPGS mRNA and FPGS enzyme activities. Moreover, cellular FPGS enzyme activities may vary by 2-3 orders of magnitude, being highest in highly proliferative cancer cells (PMID: 1435744).”

To address this comment, we performed an additional Western-blot analysis of FPGS expression in the P6T organoids overexpressing EV, FPGS or GGH, used for our drug sensitivity assays in Figure 5. Western-blot analysis shows that, next to the transcriptional level, P6T organoids are FPGS-deficient at the protein level and that our overexpression approach increases (or restores) FPGS protein expression (**new Supplementary Figure 9B**). We also included RNA-sequencing data (**new Supplementary Figure 7A**, published with the original organoid biobank by Van de Wetering *et al.*, 2015) to show that, compared to other tumor and normal organoids, P6T organoids are FPGS-deficient at the transcriptional level. These results, combined with our Western-blot analysis of FPGS protein expression, support our claims that P6T organoids are FPGS-deficient and that our overexpression strategy restores FPGS function. Assessment of FPGS protein expression omits the problem of poor correlation for *FPGS* mRNA and catalytic activity reported by Stark *et al.* (2009). For clarity, we included a statement in the discussion section to remind readers that we assume that an increase in FPGS protein expression is accompanied by an increase in FPGS catalytic activity, which we did not assess experimentally.

“Finally, some recent novel functions of FPGS were described that may also be of relevance for this study (PMID: 33676037, 36721160).”

We thank the reviewer for this suggestion. We now refer to and discuss the results of these papers in the discussion section.

July 31, 2023

RE: Life Science Alliance Manuscript #LSA-2023-02058-TR

Prof. Madelon M. Maurice
University Medical Center Utrecht
Center for Molecular Medicine and Oncode Institute
Heidelberglaan 100
Utrecht 3584CX
Netherlands

Dear Dr. Maurice,

Thank you for submitting your revised manuscript entitled "A novel antifolate suppresses growth of FPGS-deficient cells and overcomes methotrexate resistance". We would be happy to publish your paper in Life Science Alliance pending final revisions necessary to meet our formatting guidelines.

- please consult our manuscript preparation guidelines <https://www.life-science-alliance.org/manuscript-prep> and make sure your manuscript sections are in the correct order
- please add ORCID ID for corresponding (and secondary corresponding) author--you should have received instructions on how to do so
- please add a callout for Fig S8A, Fig S8B, Fig S9A, Fig S9B to your main manuscript text;
- please upload your Tables in editable .doc or excel format;
- please use the [10 author names, et al.] format in your references (i.e. limit the author names to the first 10)
- please add sizes next to blots in Figure 1E and S6

A. FINAL FILES:

B. MANUSCRIPT ORGANIZATION AND FORMATTING:

Sincerely,

Reviewer #1 (Comments to the Authors (Required)):

The authors have addressed all my concerns and suggestions. I recommend the publication of this manuscript.

Reviewer #2 (Comments to the Authors (Required)):

In their revised version of the manuscript, the authors have adequately addressed, both experimentally and in the text, all comments raised in my original review.

August 7, 2023

RE: Life Science Alliance Manuscript #LSA-2023-02058-TRR

Prof. Madelon M. Maurice
University Medical Center Utrecht
Center for Molecular Medicine and Oncode Institute
Heidelberglaan 100
Utrecht 3584CX
Netherlands

Dear Dr. Maurice,

Thank you for submitting your Research Article entitled "A novel antifolate suppresses growth of FPGS-deficient cells and overcomes methotrexate resistance". It is a pleasure to let you know that your manuscript is now accepted for publication in Life Science Alliance. Congratulations on this interesting work.

DISTRIBUTION OF MATERIALS:

Again, congratulations on a very nice paper. I hope you found the review process to be constructive and are pleased with how the manuscript was handled editorially. We look forward to future exciting submissions from your lab.

Sincerely,
